# GLH/VASA helicases promote germ granule formation to ensure the fidelity of piRNA-mediated transcriptome surveillance

Wenjun Chen[1,6,7], Jordan S. Brown[1,7], Tao He[2], Wei-Sheng Wu[3], Shikui Tu[4], Zhiping Weng [5], Donglei Zhang [2] ✉ & Heng-Chi Lee [1] ✉

piRNAs function as guardians of the genome by silencing non-self nucleic acids and transposable elements in animals. Many piRNA factors are enriched in perinuclear germ granules, but whether their localization is required for piRNA biogenesis or function is not known. Here we show that GLH/VASA helicase mutants exhibit defects in forming perinuclear condensates containing PIWI and other small RNA cofactors. These mutant animals produce largely normal levels of piRNA but are defective in triggering piRNA silencing. Strikingly, while many piRNA targets are activated in GLH mutants, we observe that hundreds of endogenous genes are aberrantly silenced by piRNAs. This defect in self versus non-self recognition is also observed in other mutants where perinuclear germ granules are disrupted. Together, our results argue that perinuclear germ granules function critically to promote the fidelity of piRNA-based transcriptome surveillance in *C. elegans* and preserve self versus non-self distinction.

Argonaute proteins use their associated small RNAs as guides to regulate targets with complementary sequences[1]. The PIWI Argonaute and its associated piRNAs are conserved guardians of the animal genome that repress transposons in the germline[2–6]. A prerequisite for any defense system is the ability to distinguish non-self from self. In *C. elegans*, piRNAs trigger gene silencing of non-self RNAs through the recruitment of RNA-dependent RNA Polymerases (RdRPs) to produce WAGO Argonaute-associated 22G-RNAs that mediate transcriptional and posttranscriptional gene silencing[7,8]. While diverse PIWI/piRNAs can recognize both foreign nucleic acids and germline-expressed "self" mRNAs[9,10], self RNAs are protected from piRNA silencing by Argonaute CSR-1 and its associated 22G-RNAs[11–13].

Intriguingly, the PIWI-related PRG-1, WAGO-1, and CSR-1 Argonautes are all enriched in germ granules, also known as P granules in *C. elegans*[4,7,14,15]. Germ (P) granules are phase-separated liquid droplets that are found in the germ cells of all animals[16,17]. The localization and formation of P granules are tightly controlled during *C. elegans* development[18,19]. In early embryos, P granules are cytoplasmic and sort to daughter cells of the germ cell lineage. As zygotic transcription begins later in embryogenesis, P granules relocalize to the nuclear periphery and remain perinuclear throughout most of germline development. Mutations that affect the formation of cytoplasmic P granules in embryos, such as mutants for germ plasm factors *meg-3 meg-4*, impact the potency and inheritance of RNAi[20–22]. However, *meg-3 meg-4* mutant adults have normal perinuclear P granules. The role perinuclear P granules play in small RNA-mediated gene regulation is less clear. Several lines of evidence suggest that perinuclear P granules may be the site of mRNA surveillance by small RNAs. First, perinuclear P granules have been shown to be the major sites of mRNA transport in germline

[1]Department of Molecular Genetics and Cell Biology, University of Chicago, Chicago, IL 60637, USA. [2]Cell Architecture Research Institute, Department of Biochemistry and Molecular Biology, School of Basic Medicine, Tongji Medical College, Huazhong University of Science and Technology, Wuhan, Hubei, China. [3]Department of Electrical Engineering, National Cheng Kung University, Tainan 701, Taiwan. [4]Department of Computer Science and Engineering, Shanghai Jiao Tong University, Shanghai, China. [5]Department of Biochemistry and Molecular Biotechnology, University of Mass. Chan Medical School, Worcester, MA 01605, USA. [6]Present address: Department of Laboratory Medicine, Third Affiliated Hospital of Sun Yat-Sen University, Guangzhou, China. [7]These authors contributed equally: Wenjun Chen, Jordan S. Brown. ✉e-mail: zhang_donglei@hust.edu.cn; hengchilee@uchicago.edu

nuclei[23]. Second, the size of perinuclear P granules shrink soon after inhibition of mRNA transcription or mRNA export[23,24], consistent with the model that newly exported mRNAs gather in perinuclear P granules. Third, a recent report showed that tethering an mRNA to P granule component PGL-1 leads to its silencing[25], suggesting the accumulation of mRNA in P granules can trigger silencing by small RNA pathways. In addition, the enzymes required for piRNA 3′-end trimming (PARN-1) and the biogenesis of CSR-1 and WAGO-associated 22G-RNAs (EGO-1) are both enriched in perinuclear P granules[26,27]. Together, these observations raise the possibility that the production and/or function of small RNAs may require the enrichment of these factors in perinuclear P granules.

The VASA-homolog RNA helicases GLH-1 and GLH-4 play a critical role in the formation of both cytoplasmic and perinuclear P granules[24,28–31]. In addition, several other P granule factors, including DEPS-1 and PGL-1, have been reported to promote both cytoplasmic and perinuclear P granule assembly[30,32]. These observations make them possible candidates as arbiters for examining P granule function in small RNA-mediated gene silencing. Here we show that GLH-1 and GLH-4 play a global role in promoting the liquid condensation of Argonautes and other small RNA factors at perinuclear foci. In addition, we find that the biogenesis of neither piRNAs nor secondary small RNAs, including WAGO or CSR-1 associated 22G-RNAs, broadly require GLH/VASA. In GLH and in other mutants with defects in forming perinuclear P granules, many piRNA targets are activated, with fewer secondary WAGO-22G-RNAs produced at piRNA targeting sites. Additionally, many functional endogenous mRNAs are aberrantly silenced by piRNAs. Together, our results suggest that GLH/VASA helicases and perinuclear P granules are critical for ensuring the fidelity of mRNA surveillance by piRNAs and that without P granules, small RNA pathways can no longer robustly identify mRNAs as self or non-self.

## Results

### GLH/VASA promotes the condensation of piRNA pathway factors

While GLH-1 plays a critical role in controlling the perinuclear localization of PIWI PRG-1[24,31], we failed to detect PRG-1 in GLH-1 complexes by mass spectrometry under native conditions[24]. We hypothesized that their interaction could be transient and therefore applied the chemical crosslinking reagent dithio-bis-maleimidoethane (DTME) to capture potentially transient interactions[33]. Indeed, using DTME-crosslinked worms, we are able to detect PRG-1 in the GLH-1 complex (Fig. 1a). In addition, we observed many other small RNA components in the GLH-1 complex, including P granule factors DEPS-1, WAGO-1, CSR-1, and Z granule factor WAGO-4 (Fig. 1a and Supplementary Data 1). These findings are consistent with previously reported GLH-1 mass spectrometry results[31]. Z granules are derived from P granules during embryogenesis and remain adjacent to P granules in the adult germline[34,35]. Similarly, an interaction between P granule and Z granule factors has previously been shown by PRG-1 mass spectrometry; PRG-1 itself interacts with factors found in both P granules and Z granules[36]. Together with the previously reported results[31], these observations raise the possibility that GLH helicases may play a global role in regulating the localization of small RNA pathway components. As VASA-like helicases GLH-1 and GLH-4 function redundantly to promote the localization of PIWI PRG-1[24], we examined the localization of various small RNA machinery in the *glh-1 glh-4* mutant. Consistent with previous findings, we found that for PIWI PRG-1, both perinuclear and cytoplasmic foci are greatly reduced in the *glh-1 glh-4* double null mutant (Fig. 1b and Supplementary Fig. 1a). For several other small RNA factors, including P granule factors DEPS-1, CSR-1, and EGO-1, and Z granule factors - WAGO-4 and ZFNX-1, their perinuclear localization is also significantly reduced in the *glh-1 glh-4* double mutant, although residual foci of some of these factors can

still be observed, specifically for CSR-1 where the overall level of perinuclear CSR-1 is reduced while the number of CSR-1 foci is similar to that in wild type animals (Fig. 1a, c and Supplementary Fig. 1a, b). We then examined the localization of MUT-16, a key factor in the assembly of Mutator granules[27]. Mutator granules house the small RNA components involved in producing WAGO-associated 22G-RNAs. Mutator granules are frequently found in close contact with perinuclear P granules[37]. Previous experiments using RNAi knockdown of *glh-1* and *glh-4* did not lead to disruption of MUT-16 localization[27]. The localization of MUT-16 was significantly disrupted in *glh-1 glh-4* double mutants (Fig. 1d and Supplementary Fig. 1c). This is consistent with a recent report demonstrating MUT-16 disruption upon simultaneous RNAi treatment against *glh-1*, *glh-4*, *pgl-1*, and *pgl-3*[38]. Together, these results show that GLH/VASA helicases play a global role in enriching small RNA machinery into the distinct liquid condensates observed throughout *C. elegans* germline development, including cytoplasmic and perinuclear P granules, Z granules and Mutator granules.

### GLH/VASA helicase mutants exhibit defects in piRNA silencing

To investigate whether GLH-1 and GLH-4 helicases are required for piRNA-mediated gene silencing, we examined whether the silencing of a piRNA reporter[13] requires these GLH/VASA helicases. This piRNA reporter is silenced in wild-type animals but is activated in the *prg-1* mutant background (Fig. 2a). Similarly, we found that the piRNA reporter is activated in the *glh-1 glh-4* double mutant background (Fig. 2a), suggesting GLH-1 and GLH-4 play a role in piRNA silencing. In addition, we found that the piRNA reporter is also activated in the *glh-1 DQAD* mutant background. Previous studies have shown that in *glh-1 DQAD* mutants, P granule factors form large aggregates and exhibit defects in the distribution of these granules in the early embryo[24,31]. Indeed, in the *glh-1 DQAD* mutant, large PRG-1 and WAGO-4 aggregates are found in the cytoplasm with a significant reduction in perinuclear PRG-1 and WAGO-4 foci (Fig. 2b and Supplementary Fig. 2a, b). In addition, these abnormal, cytoplasmic aggregates are not properly sorted to the germ cell lineage and/or not properly degraded in somatic lineages, leading to the presence of these foci in somatic lineages (Fig. 2b). Together, these data show that GLH mutants are defective in piRNA silencing.

We then wanted to understand why piRNA silencing is defective in GLH/VASA mutants. We found no significant change in expression of the GFP-targeting piRNA in these GLH/VASA mutants compared to wild-type animals (Fig. 2c). In contrast, we observed a pronounced reduction in the 22G-RNAs produced around the GFP-targeting piRNA binding site in both the *glh-1 glh-4* double and *glh-1 DQAD* mutants (Fig. 2d). These analyses suggest that GLH-1 and GLH-4 are not required for the biogenesis of the GFP-targeting piRNA, but rather promote the production of 22G-RNAs at the piRNA targeting site.

Since perinuclear P granules have been shown to be the major sites of mRNA export[23], we wondered whether GLH/VASA may also contribute to the localization of target mRNAs. We, therefore, examined the localization of GFP mRNAs, the piRNA target of our piRNA reporter, using single-molecule fluorescent in situ hybridization (smFISH). In our piRNA reporter strain where GFP is silenced by a GFP-targeting piRNA, we observed large GFP perinuclear foci and little cytoplasmic GFP signal (Fig. 2e). In the *glh-1 glh-4* double mutant piRNA reporter strain where GFP is activated, the perinuclear GFP foci were greatly reduced, while more cytoplasmic signal was observed (Fig. 2e). To quantify the extent of silenced *gfp* mRNA colocalization with P granules, we measured the amount of colocalization of *gfp* mRNA signal with PRG-1 foci. Because PRG-1 perinuclear accumulation is reduced in *glh-1 glh-4* mutants, we also analyzed a piRNA reporter strain without the piRNA that triggers *gfp* silencing. We saw that compared to the wild type, non-silenced

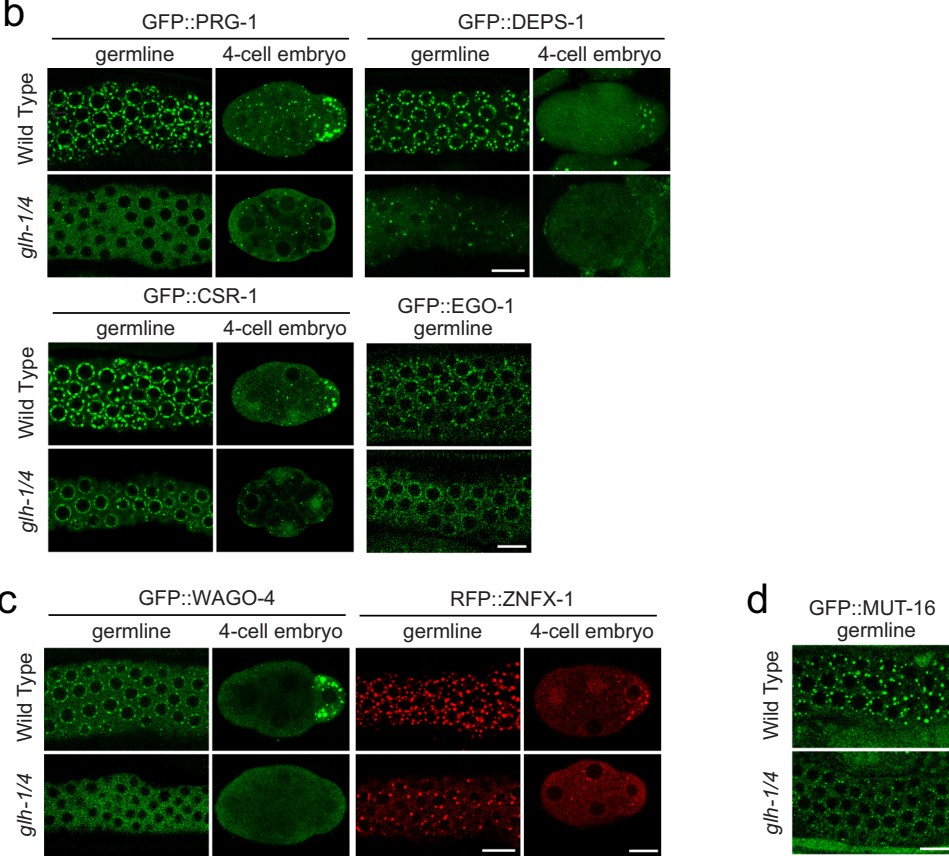

**Fig. 1 | The GLH/VASA helicases GLH-1 and GLH-4 promote the localization of germ granule factors. a** Proteomic analyses of GLH-1 complex. The numbers of peptides identified in two independent pull-down experiments are shown. **b** Fluorescent micrographs show the localization of the specific proteins known to be enriched in P granules in the wild type or the *glh-1 glh-4* mutant in the adult germline (left) and in the four-cell embryo (right). Bar, 10 micrometers.

**c** Fluorescent micrographs show the localization of the specific proteins known to be enriched in Z granules in the wild type or the *glh-1 glh-4* mutant in the adult germline (left) and in the four-cell embryo (right). Bar, 10 micrometers. **d** Fluorescent micrographs show the localization of the specific proteins known to be enriched in mutator granules in the wild type or the *glh-1 glh-4* mutant in the adult germline. Bar, 10 micrometers.

reporter and the *glh-1 glh-4* mutant reporter, the wild-type silenced reporter showed significantly more *gfp* – PRG-1 colocalization (Supplementary Fig. 2c). We did not observe this increase in colocalization when we monitored an mRNA that does not become silenced in the reporter, *nos-3*. Although this effect was significant, the extent of the colocalization was modest. This could be due to silenced *gfp* mRNA also accumulating in Mutator foci or Z granules. Consistent with this hypothesis, we noticed that many instances of *gfp* – PRG-1 colocalization were only partial, which could be indicative of overlap with granules adjacent to P granules. Therefore, GLH-1 and GLH-4 promotes the location of *gfp* mRNA at perinuclear foci in the piRNA reporter. These observations are consistent with the model that GLH/VASA promotes the accumulation of PRG-1, piRNA cofactors, and target RNAs at perinuclear foci to trigger piRNA silencing.

## P granules promote piRNA silencing

The GLH mutants examined above are defective in the localization of small RNA machinery in both perinuclear and cytoplasmic P granules. To examine whether piRNA silencing capability correlates with the ability to form either type of liquid condensate, we characterized additional genetic mutants that have been reported to show defects in forming either cytoplasmic and/or perinuclear P granules. The P granule component DEPS-1 has been shown to promote the assembly of perinuclear and cytoplasmic P granules[30]. Indeed, the formation of both perinuclear and cytoplasmic WAGO-4 condensates are compromised in *deps-1* mutants (Fig. 3a). However, we found *deps-1* mutants exhibit a reduced number of perinuclear PRG-1 condensates but retained normal cytoplasmic PRG-1 condensates. While we have not yet confirmed this finding using our piRNA reporter, a recent study reports that DEPS-1 is required for silencing of a piRNA reporter[39]. In

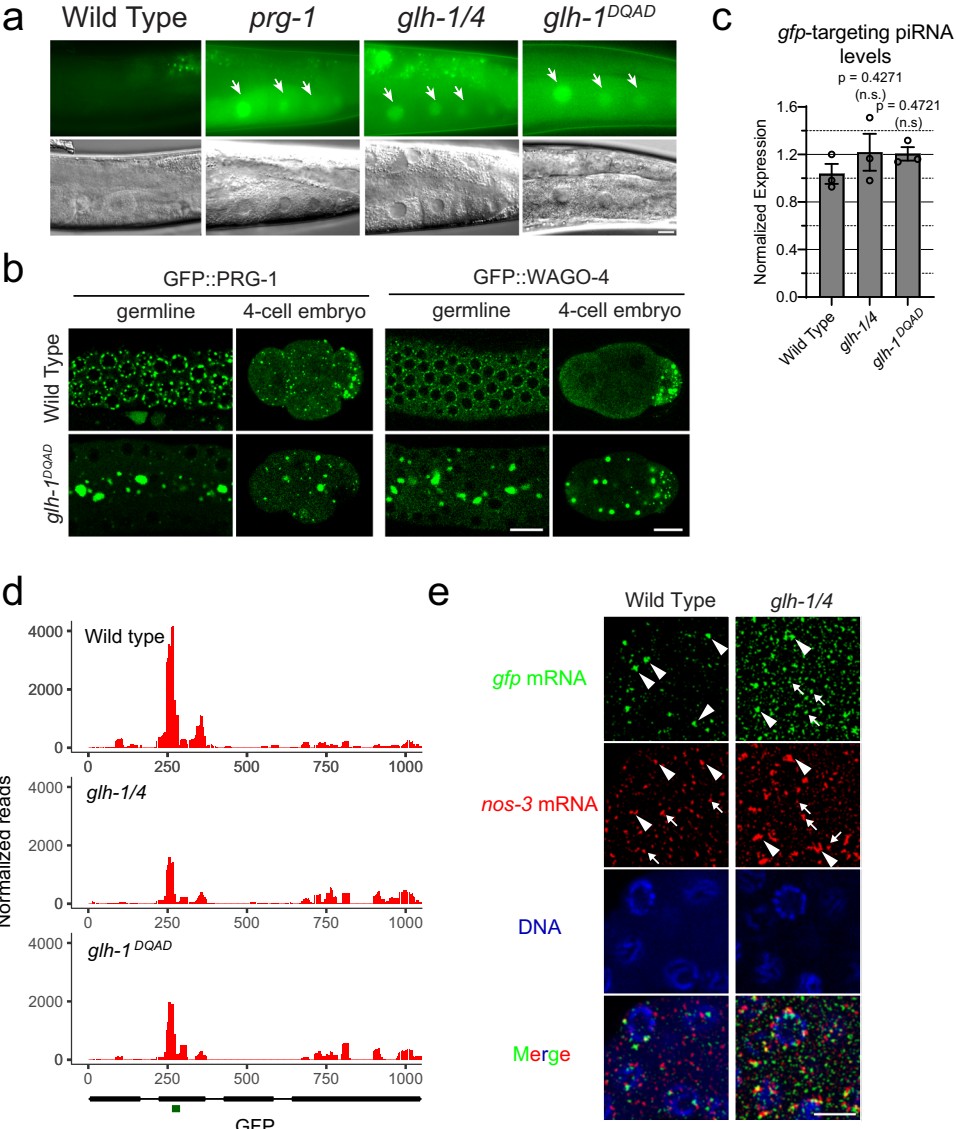

**Fig. 2 | GLH/VASA helicases are required for piRNA-dependent silencing of a piRNA reporter. a** GFP expression of the piRNA reporter in the indicated strains. Arrows indicate the adult germline nuclei in maturing oocytes with expression of GFP transgenes (top). DIC images of the corresponding reporter images (bottom). Bar, 10 micrometers. **b** Fluorescent micrographs show the localization of PRG-1 and WAGO-4 in the indicated strains in the adult germline (left) and in the four-cell embryo (right). Bar, 10 micrometers. **c** q-RT PCR measurements of the abundance of the GFP-targeting piRNA in the indicated strains. Statistical analysis was performed using a two-tailed Student's *t* test. Bars indicate the mean, errors bars indicate the standard deviation, and data points indicate values for the three technical replicates from each genotype. **d** 22G-RNAs distribution at GFP coding sequences in the indicated strains. The green bar (below) indicates the location of the GFP sequence complementary to the GFP-targeting piRNA. **e** Photomicrographs of pachytene nuclei in fixed adult gonads hybridized with single molecule fluorescent (smFISH) probes complementary to *gfp* mRNA (Green) and *nos-3* mRNA (Red) in the indicated strains. Nuclear DNA was stained with DAPI (Blue). The arrowheads indicate perinuclear mRNA foci while arrows indicate cytoplasmic mRNA foci. Bar, 5 micrometers. Photomicrographs are representative of data collected across three independent experiments. At least 10 gonads from each experiment were observed at random.

addition, we have recently shown that the N terminal phenylalanine-glycine-glycine (FGG) repeats of GLH-1 promote its perinuclear localization, leading to the recruitment of PIWI PRG-1 at perinuclear P granules[24]. Indeed, we confirmed that PRG-1 and WAGO-4 perinuclear condensates, but not their cytoplasmic condensates, are partially disrupted in the *glh-1* FGGΔ *glh-4* mutant (Fig. 3a). We also found that the piRNA reporter is activated in this *glh-1 FGGΔ glh-4* strain (Fig. 3b). These results indicate that mutants defective in forming perinuclear P granules exhibit defects in piRNA silencing. We then examined the *meg-3 meg-4* mutant, in which the localization of the small RNA machinery is disrupted in cytoplasmic P granules found in the early embryos but still exhibit wild type like adult perinuclear P granules[22,40] (Fig. 3a). We found that the piRNA reporter is also activated in ~30

percent of *meg-3 meg-4* mutant animals (Fig. 3b). Together, these results indicate that the localization of piRNA factors at adult perinuclear P granules and embryonic cytoplasmic P granules both contribute to their function in piRNA silencing.

**Perinuclear P granules promote initiation of piRNA silencing**
Our observations that GLH/VASA promotes the perinuclear localization of piRNA factors and their target mRNAs raises the possibility that their enrichment in P granules allows PIWI PRG-1 and its cofactors to efficiently identify their targets and to trigger gene silencing. To test whether GLH/VASA mutants exhibit defects in de novo piRNA-mediated gene silencing, we microinjected a synthetic piRNA-expressing plasmid into transgenic worms expressing a silencing-

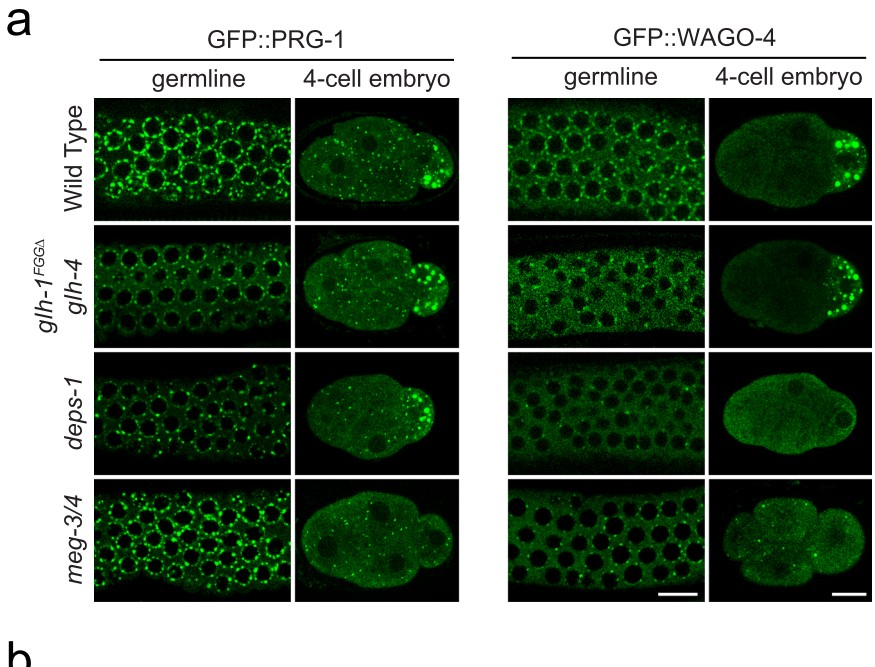

Fig. 3 | **Mutants with defective in either perinuclear or cytoplasmic P granules exhibit defects in gene silencing. a** Fluorescent micrographs show the localization of PRG-1 and WAGO-4 in the indicated strains in the adult germline (left) and in the four-cell embryo (right). Bar, 10 micrometers. Quantification of each genotype's effect on granule properties is provided in Supplementary Fig. 2a. **b** GFP expression in the piRNA reporter and the ability to form perinuclear or cytoplasmic PRG-1 granules in the indicated strains.

| Strain Genotype | Perinuclear PRG-1 granules (adult) | Cytoplasmic PRG-1 granules (embryo) | piRNA sensor expression (%) |
|---|---|---|---|
| Wild Type | +++ | +++ | 2 |
| *prg-1* | - | - | 100 |
| *glh-1/4* | - | - | 100 |
| *glh-1$^{DQAD}$* | +* | +* | 95 |
| *glh-1$^{FGGΔ}$ glh-4* | ++ | +++ | 95 |
| *deps-1* | ++ | +++ | N/A |
| *meg-3/4* | +++ | - | 30 |

n>50

prone GFP::CDK-1 transgene[10] and monitored the silencing of the GFP transgene by the GFP-targeting piRNA. While 97% of the injected wild-type animals successfully triggered silencing of the GFP transgene by the GFP-targeting piRNA by the F2 generation, only 5% of the *glh-1 glh-4* double mutant animals and 36% of the *glh-1 DQAD* mutant animals triggered silencing of the GFP transgene (Supplementary Fig. 3a). In addition, we observed that only 40% of the *deps-1* mutant animals and 31% of the *glh-1 FGGΔ glh-4* mutant animals triggered silencing of the GFP transgene. In contrast, *meg-3 meg-4* mutants exhibit a normal, wild type ability to trigger silencing of the GFP transgene by the GFP-targeting piRNA. Taken together, these assays show that mutants defective in forming perinuclear P granules are compromised in their ability to initiate piRNA silencing.

**Perinuclear P granules promote silencing of piRNA targets**
Our analyses of the piRNA reporter suggest that GLH/VASA is not required for the biogenesis of a GFP-targeting piRNA (Fig. 2c). Consistent with the reporter analyses, we observed slightly reduced or normal piRNA levels in the *glh-1 glh-4* double mutant or in *glh-1 DQAD* mutant, respectively (Supplementary Fig. 4a). In addition, we found that the length of piRNAs was not changed in the *glh-1 glh-4* double

mutant or in the *glh-1 DQAD* mutant (Supplementary Fig. 4b), suggesting that the 3′ processing of piRNAs by PARN-1 is not affected. On the other hand, over 41% of WAGO targets (genes known to be silenced by WAGO-22G-RNAs) exhibit a two-fold decrease in 22G-RNAs in the *glh-1 glh-4* double mutant and over 51% in the *glh-1 DQAD* mutant (Fig. 4a, left and Supplementary Fig. 4c)[7]. To confirm that the reduction in 22G-RNAs is due to decreased production of WAGO-associated 22G-RNAs, we performed an immunoprecipitation with HRDE-1 (WAGO-9) and compared wild type HRDE-1 22G-RNAs to those in *glh-1* mutants, which exhibit a partial defect in forming perinuclear PRG-1 foci[24]. (Fig. 4a, right). We used *glh-1* single mutants to perform the HRDE-1 IP experiment due to technical limitations, as it is difficult to accumulate enough material for immunoprecipitation using the nearly sterile *glh-1 glh-4* double mutants. The *glh-1* single mutant has a milder defect in WAGO-22G-RNA accumulation than the *glh-1 glh-4* mutant: 33% of WAGO targets exhibit at least a 2-fold decrease in total 22G-RNAs in the *glh-1* mutant compared to 41% in the *glh-1 glh-4* (Supplementary Fig. 4d and Fig. 4a). Our HRDE-1 IP experiment revealed that 48% of WAGO targets exhibit a 2-fold decrease in HRDE-1 22G-RNAs in the *glh-1* mutant. To obtain a more conservative list of affected WAGO genes, we also applied a statistical threshold and observed that 16% of WAGO

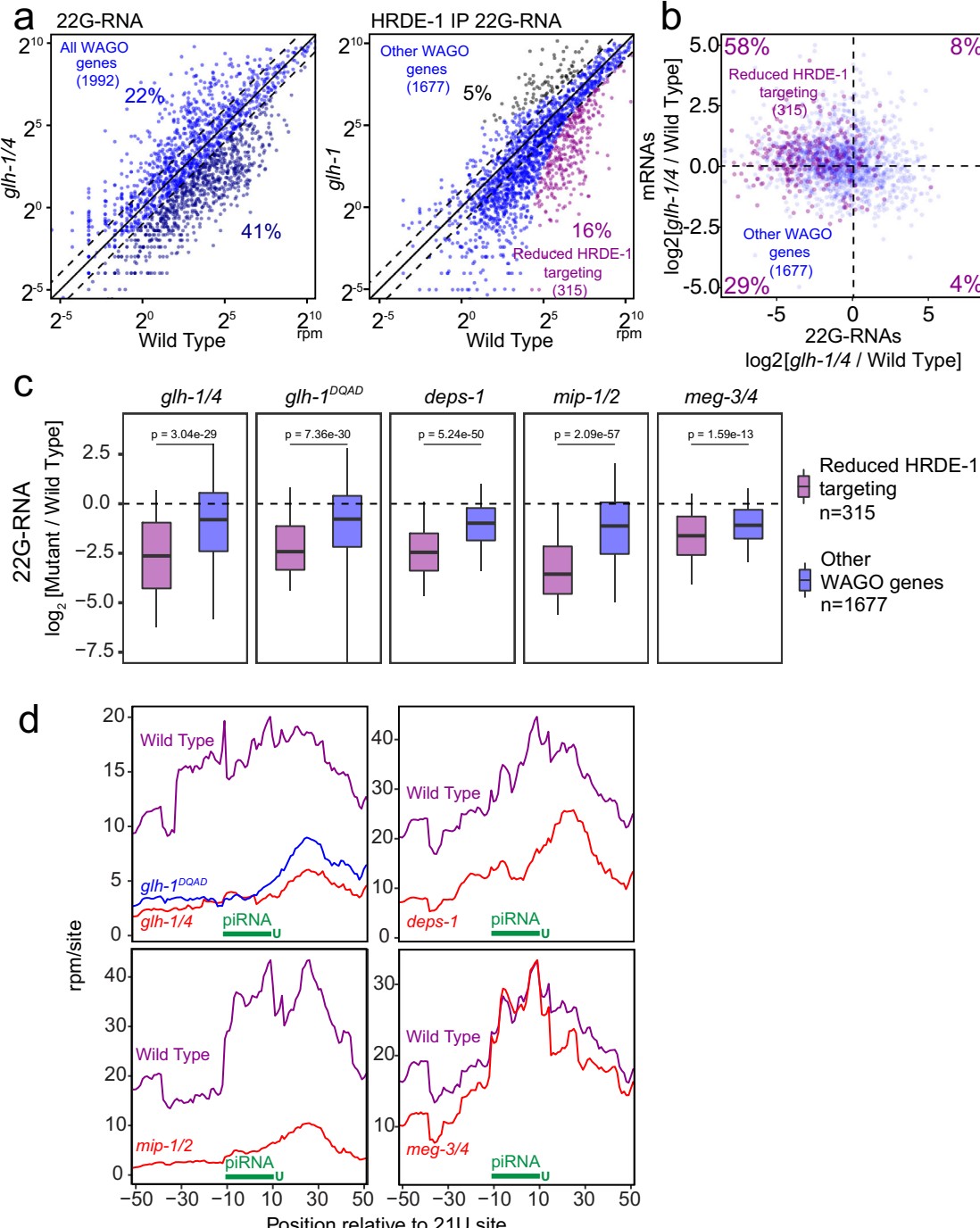

**Fig. 4 | GLH/VASA mutants exhibit defects in producing secondary WAGO-22G-RNAs. a** A scatter plot showing the abundance of all 22G-RNAs (left) or HRDE-1 bound 22G-RNAs (right) mapped to each WAGO target in wild-type worms compared to indicated strains. (Left) The percentage of WAGO targets with 2-fold increased or decreased 22G-RNAs in mutants are shown. (Right) The percentage of significantly changed (adjusted $P < 0.05$ [see Methods for details] and 2-fold) WAGO targets are shown. The three diagonal lines indicate a two-fold increase (top), no change (middle), or a two-fold depletion (bottom) in the indicated mutant strains. **b** A scatter plot showing the mRNA vs 22G-RNA $\log_2$ expression changes for WAGO targets in *glh-1 glh-4* mutants vs wild type worms. The upper left quadrant corresponds to WAGO targets that have become activated in the mutant (increased mRNAs and decreased 22G-RNAs). Percentages are shown in each quadrant to indicate the proportion of WAGO targets with reduced HRDE-1-associated 22G-RNAs in the *glh-1* mutant that fall in that quadrant. **c** 22G-RNA changes for WAGO targets with reduced HRDE-1-associated 22G-RNAs in the *glh-1* mutant versus all other WAGO targets in the indicated strains versus wild type worms. Statistical analysis was performed using a two-tailed Mann–Whitney Wilcoxon test. For all boxplots, lines display median values, boxes display first and third quartiles, and whiskers display 5th and 95th percentiles. **d** Density of 22G-RNAs within a 100-nt window around predicted piRNA target sites in the indicated strains. Computed by summing 22G-RNA density per piRNA targeting site in WAGO targets with reduced HRDE-1 associated 22G-RNAs in the *glh-1* mutant. The plots are centered on the 10th nucleotide of piRNAs.

genes were significantly reduced in HRDE-1 22G-RNA accumulation in *glh-1* mutants, compared to only 5% of WAGO genes significantly increased. Using mRNA sequencing, we found the majority of genes exhibiting significantly reduced HRDE-1 22G-RNA levels in *glh-1* mutants had increased mRNA levels in *glh-1 glh-4* double mutants and in the *glh-1 DQAD* mutants (Fig. 4b and Supplementary Fig. 4e). Several transposons were among those shared activated genes, including TC2 and PAL8C_1 (Supplementary Fig. 4f). These observations indicate that GLH/VASA mutants exhibit defects in silencing WAGO targets. As the levels of 22 G RNAs from WAGO targets were reported to be reduced in *meg-3 meg-4* double mutants[22] which exhibit defects only in cytoplasmic but not in perinuclear PRG-1 granules, we wondered whether distinct or common WAGO targets exhibit 22G-RNA defects in mutants defective in cytoplasmic versus perinuclear P granules. We noticed that those WAGO genes with significantly reduced HRDE-1 22G-RNAs in *glh-1* mutants exhibit a greater reduction in 22G-RNAs compared to other WAGO target genes in the *glh-1 glh-4* mutants and in *glh-1* DQAD mutants. A similar trend of 22G-RNA changes is also found in the *deps-1* mutant and in the *mip-1; mip-2* double mutant, which has also recently been shown to play an important role in P granule assembly[41]. In *meg-3 meg-4* mutants, which exhibit defects in the formation of embryonic cytoplasmic P granules, we see much less difference in 22G-RNA between these WAGO genes, suggesting WAGO-22G-RNA biogenesis is globally compromised as previously reported (Fig. 4c)[22]. These results suggest that hundreds WAGO target genes are preferentially activated in GLH/VASA mutants and other mutants exhibiting defects in perinuclear P granules.

Since our piRNA reporter analyses suggest that GLH/VASA is critical for the production of downstream 22G-RNAs around piRNA targeting sites, we wondered whether the production of piRNA-dependent 22G-RNAs are preferentially affected in GLH/VASA mutants and other mutants exhibiting defects in forming perinuclear P granules[42]. Indeed, in *glh-1 glh-4*, *glh-1* DQAD and *deps-1* mutants, those WAGO target genes in which the production of 22G-RNAs most depends on PRG-1 exhibit greater reductions in 22G-RNAs than other WAGO targets (Supplementary Fig. 4g). A still significant but lesser difference in 22G-RNA production was found for *meg-3 meg-4* mutants (Supplementary Fig. 4g). To specifically examine the production of 22G-RNA production at piRNA targeting sites, we compared the local production of 22G-RNAs at predicted piRNA target sites[10,43]. For those WAGO targets with significantly reduced HRDE-1 22G-RNAs in *glh-1* mutants, 22G-RNAs are enriched around predicted piRNA target sites in wild type but much less so in *glh-1 glh-4*, *glh-1 DQAD*, or *deps-1* mutants (Fig. 4d). In contrast, while 22G-RNAs are overall slightly reduced in *meg-3 meg-4* mutants, 22G-RNAs remain enriched around these predicted piRNA target sites (Fig. 4e). Similar results were observed when we looked at predicted piRNA target sites in all WAGO genes (Supplementary Fig. 4h). These observations suggest that the production of piRNA-dependent 22G-RNAs is preferentially compromised in the *glh* and *deps-1* mutants. Together, these observations suggest that piRNA-dependent WAGO target genes are preferentially activated in VASA mutants and other mutants exhibiting defects in forming perinuclear P granules.

### Perinuclear P granules prevent silencing of self genes

Essentially all germline transcripts are targeted either by 22G-RNAs associated with WAGO or CSR-1 Argonautes, which can silence or license the expression of their targets, respectively[7,11,12]. Since CSR-1 is present in GLH-1 complexes and the formation of CSR-1 perinuclear condensates is promoted by GLHs (Fig. 1a, b), we wondered whether GLHs may regulate the biogenesis and/or function of CSR-1-associated 22G-RNAs. It has recently been shown that simultaneous RNAi against four P granule factors (*glh-1*, *glh-4*, *pgl-1*, and *pgl-3*) does not lead to global changes to 22G-RNAs antisense to CSR-1 targeted genes[38]. Consistent with this finding, we found that 22G-RNAs antisense to CSR-

1 target genes remain mostly unchanged in *glh* mutants compared to wild type (Fig. 5a, left and Supplementary Fig. 5a). Intriguingly, in *glh-1 glh-4* double mutants, we noticed that 717 CSR-1 genes (15%) exhibit at least a two-fold increase in 22G-RNAs that mapped to them. Similar to the analyses of WAGO targets, we performed HRDE-1 IP to ask whether these changes were due to increased production of WAGO-associated 22G-RNAs. Again, we used the more fertile but less severely affected *glh-1* single mutants to perform the IP, which from total 22G-RNA sequencing show 330 CSR-1 targets (7%) with two-fold increased 22G-RNA accumulation compared to the 717 CSR-1 targets (15%) upregulated in double mutants (Supplementary Fig. 5b and Fig. 5a). We again applied a statistical threshold to our IP data to obtain a more conservative list of affected CSR-1 genes. Strikingly, HRDE-1 (WAGO-9) immunoprecipitation analyses of the *glh-1* mutant showed that 320 genes were significantly enhanced in HRDE-1 22G-RNA accumulation in *glh-1* mutants (Fig. 5a). mRNA sequencing analyses confirmed an overall reduction in mRNA levels for these enhanced HRDE-1 targeted genes in *glh-1 glh-4* double and in *glh-1 DQAD* mutants (Fig. 5b and Supplementary Fig. 5a). The aberrantly silenced CSR-1 targets include several functional genes involved in germline development, including *nos-2*, *puf-6* and *puf-7*.

We wondered whether the aberrant silencing of CSR-1 genes occurs in other mutants defective in cytoplasmic and/or perinuclear P granule formation.

In *glh-1 DQAD* mutants, a significant increase in 22G-RNAs and decrease in mRNAs are found for those CSR-1 genes with enhanced HRDE-1 22G-RNA levels in *glh-1* mutants. Similar changes of 22G-RNA or mRNA levels are found in other mutants exhibiting defects in the formation of perinuclear P granules, including *deps-1* and *mip-1/2* mutants. In contrast, in the *meg-3 meg-4* mutant that only disrupts the formation of embryonic cytoplasmic P granules in embryos, we did not observe an increase in 22G-RNAs nor a decrease in mRNAs for these CSR-1 genes with enhanced HRDE-1 22G-RNA levels in *glh-1* mutants (Fig. 5c). These observations suggest that in mutants with abnormal perinuclear P granules, a group of CSR-1 target genes are aberrantly silenced by WAGO-22G-RNAs.

### PRG-1 is required for the aberrant self silencing

Because perinuclear P granule loss affects the localization but not abundance of PIWI PRG-1[24], we wondered whether PRG-1 may be aberrantly recognizing these CSR-1 genes when perinuclear P granule formation is compromised. Indeed, we found that the aberrant production of 22G-RNAs against CSR-1 genes with enhanced HRDE-1 22G-RNA levels in *glh-1* mutants are significantly reduced in *glh-1 glh-4 prg-1* triple mutants compared to *glh-1 glh-4* double mutants (Fig. 5d). The mutation of *prg-1* suppresses the aberrant accumulation of CSR-1 22G-RNAs in *glh-1 glh-4* double mutants by 2-fold or more for nearly two-thirds of affected genes (Supplementary Fig. 5c). These data suggest that most of the aberrant mRNA silencing of CSR-1 genes found in GLH/VASA mutants is triggered by piRNAs. Together, these data suggest VASA not only promotes silencing of non-self (WAGO targets), but also promotes licensing of self (CSR-1 targets) to avoid silencing by piRNAs. Previous studies have shown that improper silencing of CSR-1 targets can occur in *prg-1* mutants[44,45]. In contrast, our observations demonstrate that aberrant silencing of CSR-1 targets can also occur in the presence of PRG-1. In fact, here we showed that in VASA mutants, PRG-1 is responsible for triggering the aberrant silencing of many CSR-1 targets.

If self transcript licensing depends on P granules, then these transcripts should be present in perinuclear condensates. Using RNA smFISH analyses, we examined the localization of *ceh-49*, an aberrantly silenced mRNA normally protected from silencing by CSR-1 (Fig. 5e), and noticed that *ceh-49* mRNAs were expressed in the pachytene region of the germline in wild type animals and both perinuclear and cytoplasmic signals are detected (Fig. 5f). In *glh-1 glh-4* mutants, both

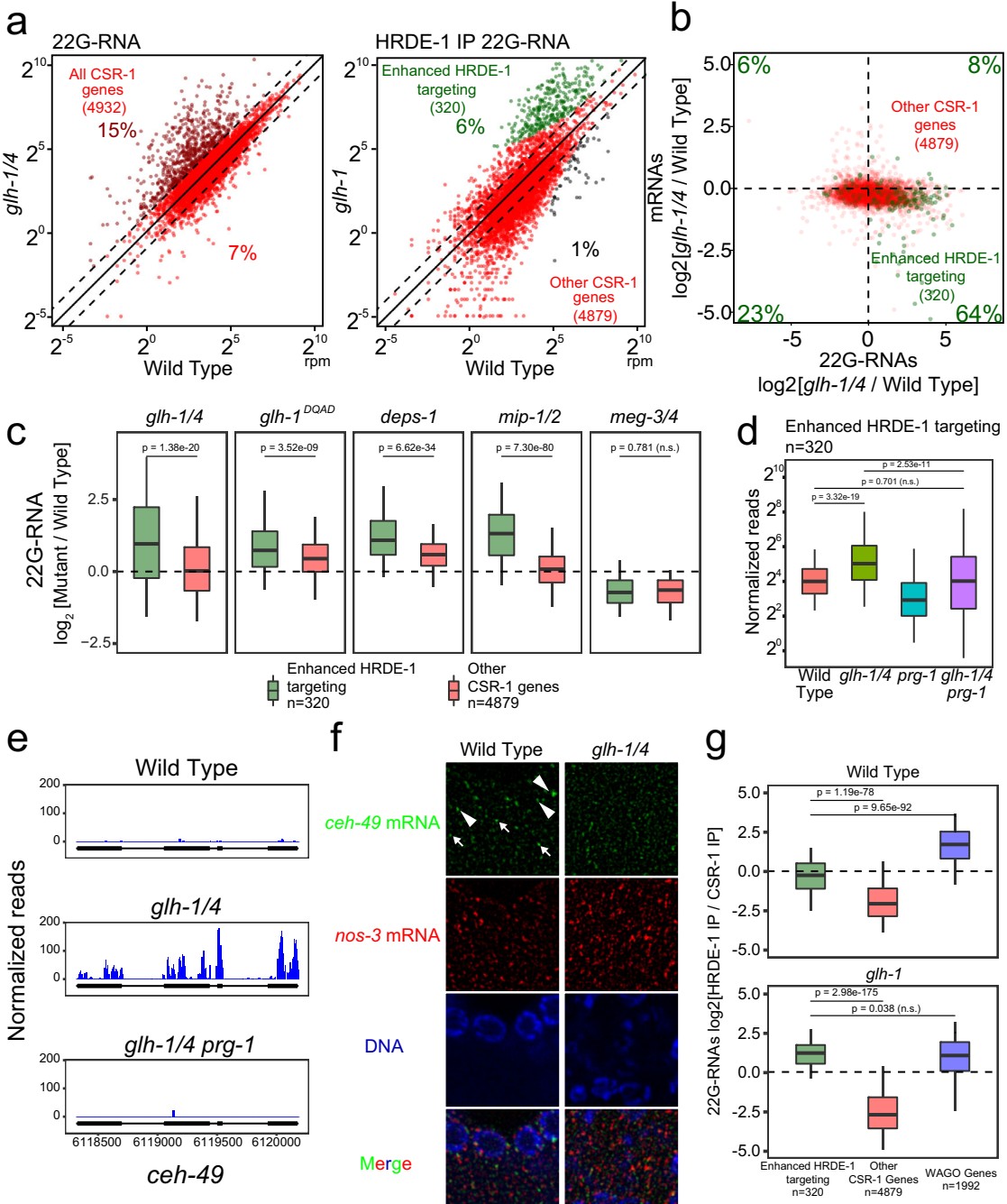

**Fig. 5 | Many functional germline genes are silenced in mutants with defects in forming perinuclear PRG-1 granules. a** A scatter plot showing the abundance of all 22G-RNAs (left) or HRDE-1 bound 22G-RNAs (right) mapped to each CSR-1 target in wild-type worms compared to indicated strains. (Left) Percentages of CSR-1 targets with twofold increased or decreased 22G-RNAs in mutants are shown. (Right) The percentage of significantly changed (adjusted *P* < 0.05 [see Methods for details] and twofold) CSR-1 targets are shown. Diagonal lines indicate a two-fold increase (top), no change (middle), or a twofold depletion (bottom) in the indicated mutant strains. **b** A scatter plot showing the mRNA vs 22G-RNA expression changes for CSR-1 targets in the indicated strain. Percentages in each quadrant indicate the proportion of CSR-1 targets with enhanced HRDE-1 targeting that fall in that quadrant. **c** 22G-RNA fold changes for CSR-1 targets with enhanced HRDE-1 targeting versus all other CSR-1 targets in the indicated strains versus wild-type worms. Statistical analysis was performed using a two-tailed Mann–Whitney Wilcoxon test. For all boxplots, lines display median values, boxes display first and third quartiles, and whiskers display 5th and 95th percentiles. **d** 22G-RNA accumulation for CSR-1

targets with enhanced HRDE-1 targeting compared between the indicated strains. Statistical analysis was performed using a two-tailed Mann–Whitney Wilcoxon test. For all boxplots, lines display median values, boxes display first and third quartiles, and whiskers display 5th and 95th percentiles. **e** 22G-RNAs distribution in the enhanced HRDE-1-associated 22G-RNA target *ceh-49* in the indicated strains. **f** Photomicrographs of pachytene nuclei in adult gonads hybridized with specified single molecule fluorescent (smFISH) probes in the indicated strains. Nuclear DNA was stained with DAPI (Blue). Arrowheads indicate perinuclear mRNA foci. Arrows indicate cytoplasmic mRNA foci. Bar, 5 micrometers. **g** 22G-RNA enrichment in HRDE-1 versus CSR-1 IP experiments in the indicated strains for CSR-1 targets and WAGO targets. Dotted line indicates no enrichment for either Argonaute. Statistical analysis was performed using a two-tailed Mann–Whitney Wilcoxon test. For all boxplots, lines display median values, boxes display first and third quartiles, and whiskers display 5th and 95th percentiles. Distributions represent data collected across two biological replicates for CSR-1 IP and a single experiment for HRDE-1 IP.

perinuclear and cytoplasmic *ceh-49* signals were decreased. We performed colocalization analysis with PRG-1 to measure the extent to which *ceh-49* accumulates in P granules compared to an mRNA that does not become aberrantly silenced by piRNAs in VASA mutants. We observed greater colocalization of *ceh-49* mRNA with PRG-1 compared to the colocalization of *nos-3* mRNA with PRG-1 (Supplementary Fig. 5d). This effect was dependent on *glh-1 glh-4*. These results are consistent with the model that the P granule localization of some mRNA transcripts is critical for their protection from piRNA silencing[22].

We wondered why some CSR-1 targeted genes gained aberrant silencing in GLH/VASA mutants. We hypothesized that these genes might already be prone to silencing in wild-type animals. To test this hypothesis, we compared the amounts of HRDE-1-associated 22G-RNAs to CSR-1-associated 22G-RNAs on these genes. Interestingly, in wild-type animals, CSR-1 targets with enhanced HRDE-1 22G-RNA levels in *glh-1* mutants already exhibited a significantly higher ratio of HRDE-1 associated 22G-RNAs to CSR-1 associated 22G-RNAs compared to other CSR-1 targets (Fig. 5g). This ratio for CSR-1 genes with enhanced HRDE-1 22G-RNA levels in *glh-1* mutants favors HRDE-1 even more in *glh-1* mutants (Fig. 5g). The HRDE-1 to CSR-1 associated small RNA ratio for this group of genes was so far shifted in the *glh-1* mutant that they now more closely resemble WAGO genes (Fig. 5g). We wondered whether CSR-1 targets which did not meet the stringent cutoff to be called enhanced HRDE-1 targeted genes but still showed two-fold increased 22G-RNA targeting in *glh-1 glh-4* mutants also showed this trend. We found that the HRDE-1/CSR-1 IP ratio for these targets showed a similar trend of significant elevation compared to other CSR-1 genes, but their ratio was not on average as elevated as the ratio for the targets with elevated HRDE-1 targeting (Supplementary Fig. 6a), suggesting that while the group of genes which show significant HRDE-1 enhanced targeting in the mutant are distinct, there are likely other genes which become mis-regulated but just to a lesser extent. These data indicate that these aberrantly silenced CSR-1 genes are already more targeted by silencing machinery than other CSR-1 genes in wild-type animals. In addition, previous studies have shown CSR-1 22G-RNAs are enriched at the 3′ end of transcripts[35,38]. However, CSR-1 22G-RNAs are more distributed across the whole gene body for these aberrantly silenced CSR-1 targets, similar to the distribution of CSR-1 22G-RNAs observed for WAGO targets (Supplementary Fig. 6b). To characterize the extent to which these features correlate with enhanced HRDE-1 targeting, we defined the overlap between CSR-1 targets which do not show 3′ end targeting enrichment, CSR-1 targets which have a high HRDE-1/CSR-1 IP ratio, and CSR-1 targets which show enhanced HRDE-1 targeting in *glh-1* mutants. We saw that over two-thirds of CSR-1 genes that gain aberrant HRDE-1 22G-RNAs in *glh-1* mutants also share either of these other two features (Supplementary Fig. 6c). We further characterized this effect by comparing the increase in HRDE-1 targeting following *glh-1* loss for groups of CSR-1 targets that have varying levels of 3′ end targeting enrichment and HRDE-1/CSR-1 IP ratios. We saw that while groups of genes with no 3′ end enrichment or with a high HRDE-1/CSR-1 IP ratio certainly tend to gain more HRDE-1 22G-RNAs in *glh-1* mutants, the magnitude of these effects are not distinct enough to fully predict which CSR-1 targets would gain HRDE-1 22G-RNAs in P granule mutants (Supplementary Fig. 6d). Together, these observations suggest that perinuclear P granules are critical for protecting hundreds of CSR-1 transcripts from aberrant piRNA silencing while at the same time promoting piRNA silencing on WAGO targets.

## Discussion

Mutations in VASA helicase genes have been reported to lead to defects in the localization of PIWI in various animals[24,31,46–48]. Whether the localization of other small RNA factors in P granules and other perinuclear granules also relies on GLH/VASA helicases had only been explored sporadically. In *C. elegans*, piRNAs and other small RNA pathway factors are concentrated in distinct but partially overlapping granules, including P granules, Z granules, and Mutator granules[27,34], suggesting these granules interact to facilitate gene regulation by small RNAs. Here we expand on these analyses to examine the roles GLH-1 and GLH-4 play in the localization of various small RNA factors, include P granule components - DEPS-1, EGO-1, CSR-1, Z granule components - WAGO-4 and ZNFX-1, and Mutator component MUT-16. We found that the formation of perinuclear condensation of these components are each compromised in *glh-1 glh-4* double mutants. There are some important caveats to our findings. First, not all the components examined are compromised to the same extend in *glh-1 glh-4* mutants. Second, it is possible that while condensates were observably disrupted upon our genetic manipulation, there could be small RNA factors that remain at the nuclear periphery that, although undetectable by microscopy, still function sufficiently to contribute self versus non-self distinction. Nevertheless, our observations suggest that GLH/VASA plays a global role in promoting the formation of these various liquid condensates.

Phase-separated condensates are capable of concentrating various proteins and RNAs, but whether these condensates indeed play a biological role remains controversial. For example, a previous study has reported that P body formation is a consequence but not a cause of miRNA-mediated gene silencing[49]. In this study, we found that the production of piRNAs and other 22G-RNAs are not grossly affected in GLH/VASA mutants and other mutants defective in forming perinuclear condensates. Instead, the small RNA-based distinction of self and non-self RNAs are compromised in these mutants. Specifically, piRNA-dependent silencing of non-self is reduced, leading to increased mRNA expression. Simultaneously, hundreds of self RNAs (CSR-1 targets) were aberrantly silenced by piRNAs, leading to reduced mRNA expression. Our results demonstrate that a significant portion of germline transcripts are misregulated by HRDE-1 22 G RNAs in VASA mutants. As both WAGO and CSR-1 22G-RNAs can establish epigenetic memories[8,50], the silenced or expressed state of many germline transcripts may be preserved in the absence of P granules. In this model, those mRNAs which did not establish robust epigenetic memories would be those that exhibit more misregulation in P granule mutants. Indeed, we observed that PRG-1-dependent 22G-RNA targets, which depend on PRG-1/piRNAs at each generation to trigger gene silencing, are those which exhibit a greater reduction of 22G-RNAs on WAGO targets in *glh-1;glh-4* mutants (Supplementary Fig. 4g). In addition, a previous study from the Ketting lab has demonstrated that re-establishment of the 22G-RNA system in *prg-1* mutants leads to gene-mis-regulation in a stochastic manner that varies between worms[44]. As our measurements of 22G-RNAs or mRNAs are from hundreds of thousands of worms, stochastic activation or silencing that may exist in individual worms may not be detected. Examining whether P granule mutant worms exhibit aberrant activation or silencing of germline transcripts stochastically will be interesting in future studies. Notably, while a similar misregulation of CSR-1 and WAGO targets can be found in several mutants defective in P granule assembly, such as VASA, *deps-1*, *pgl-1* and *mip-1; mip-2* mutants, such defects are distinct from those defects observed in mutants defective for Mutator assembly[27] or Z granule function[34,35,51]. Together, these data argue that perinuclear P granules are critical for the fidelity of piRNA-based surveillance in *C. elegans* and provide an environment that allows distinct Argonaute proteins, such as PRG-1 and CSR-1, to distinguish self from non-self. Our observations support a model where P granules act as the checkpoint for piRNA-mediated gene silencing, where the P granule promotes piRNA target recognition for non-self genes while allowing CSR-1 to guard self genes (Fig. 6). In GLH/VASA mutants, the P granule fails to form, leading to the dispersal of small RNA factors including PRG-1, WAGOs, and CSR-1 into the cytoplasm. When denied the environment of the P granule, hundreds of typically silenced non-self transcripts fail to contact silencing machinery leading to their expression while hundreds of typically expressed self transcripts fail to

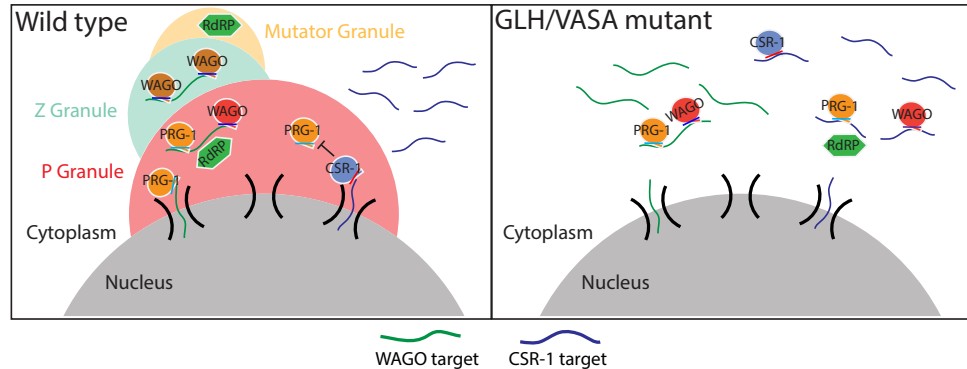

**Fig. 6 | P granules promote piRNA-targeting fidelity.** Model showing the P granule working in concert with Z and Mutator granules to maintain an environment that protects specific self genes from silencing and promotes the silencing of specific non-self genes (Left). In GLH/VASA mutants where granule integrity is lost, hundreds of typically expressed transcripts become targeted by PRG-1 and silencing machinery while hundreds of typically silenced transcripts become dispersed into the cytoplasm and expressed (Right).

contact CSR-1 leading to their repression by PRG-1. This model suggests P granules act as a specialized environment where transcripts are allowed residence time to properly contact either silencing or licensing machinery, as evidenced by our smFISH data that demonstrates the GLH/VASA-dependent perinuclear localization of both silenced and expressed mRNAs (Figs. 2e, 6f). Taken together, our study reveals the critical role of perinuclear P granules in promoting the fidelity of self and non-self nucleic acids distinction and has broad implications for the functions of other RNA-enriched liquid condensates.

## Methods

### Caenorhabditis elegans strains
Animals were grown on standard nematode growth media (NGM) plates seeded with the Escherichia coli OP50 strain at 20 °C or temperatures where indicated. Some strains were obtained from the Caenorhabditis Genetics Center (CGC). The strains used in this study are glh-1(uoc1) I, glh-1$^{DQAD}$(uoc3) I, glh-1$^{FGGΔ}$(uoc4) I, glh-1(uoc5) glh-4(gk225) I/hT2 [bli-4(e937) let-?(q782) qIs48] (I;III), deps-1(bn124) I, meg-3(ax3055) meg-4(ax3052) X., and mip-1(uae1) III; mip-2(uae2) V.

### Fluorescence microscopy and image processing
GFP- and RFP-tagged fluorescent proteins were visualized in living nematodes or dissected embryos by mounting young adult animals on 2% agarose pads with M9 buffer (22 mM KH2PO4, 42 mM Na2HPO4, and 86 mM NaCl) with 10–50 mM levamisole, or mounting one-cell embryos on 2% agarose pads by dissecting gravid hermaphrodites into egg salt buffer (5 mM HEPES pH 7.4, 118 mM NaCl, 40 mM KCl, 3.4 mM MgCl2, and 3.4 mM CaCl2). Fluorescent images were captured using a Zeiss LSM800 confocal microscope with a Plan-Apochromat ×63/1.4 Oil DIC M27 objective.

Image processing and quantification of fluorescent puncta were performed using ImageJ. Single slice images of gonads and images from maximum intensity projections of z-series of one-cell embryos were used for quantification. Regions of interest (ROIs) were selected, and areas of ROIs were measured. Image thresholds were set manually, and fluorescent puncta were selected. Puncta density was calculated as the number of fluorescent puncta in the ROI divided by the area of ROI. Integrated intensities and sizes of fluorescent puncta in all ROIs were measured. Data were analyzed by student's t test or one-way ANOVA followed by Tukey's correction for multiple comparisons.

### piRNA initiation assay
Initiation of piRNA silencing by the introduction of extrachromosomal arrays was performed as previously described[10]. Briefly, a plasmid carrying a synthetic piRNA was co-injected with pRF4 plasmid carrying the dominant injection marker rol-6dm into young adult gonads of indicated genotypes. Progeny were screened for the presence of the injection marker, and progeny of these F1s (F2 progeny of injected worms) were screened for their ability to silence the gfp::cdk-1 transgene if they were able to inherit the injection marker.

### RNA isolation and quantitative real-time PCR
Total RNA was extracted using the standard method with TRIzol reagent (Invitrogen) from whole animals of ~100,000 synchronized young adults. Stem-loop real-time PCR was performed to measure piRNA levels. 1 μg of total RNA was reverse transcribed with Super-Script IV Reverse Transcriptase (Invitrogen) in 1× reaction buffer, 5U SUPERase-In RNase Inhibitor (Invitrogen), 1 mM dNTPs, and 50 pM stem-loop reverse primer 5'-CTCAACTGGTGTCGTGGAGTCGG-CAATTCAGTTGAG-n8-3' (n8 = reverse complement sequences of last eight nucleotide acids in piRNAs). Each real-time PCR reaction consisted 4 μL of cDNA, 1 μM forward piRNA primer 5'-ACACTC-CAGCTGGG-n16-3' (n16 = first 16 nucleotide acids in piRNAs), and 1 μM universal reverse primer 5'-CTCAAGTGTCGTGGAGTCGGCAA-3'. The amplification was performed using Power SYBR Green (Applied Biosystems) on the Bio-Rad CFX96 Touch Real-Time PCR Detection System. The experiments were repeated for a total of three biological replicates.

### Small RNA sequencing
Total RNA was extracted from whole animals of ~100,000 synchronized young adults as described above. Small (<200nt) RNAs were enriched with mirVana miRNA Isolation Kit (Ambion). In brief, 80 μL (200–300 μg) of total RNA, 400 μl of mirVana lysis/binding buffer and 48 μL of mirVana homogenate buffer were mixed well and incubated at room temperature for 5 min. Then 176 μL of 100% ethanol was added and samples spun at 2500 × g for 4 min at room temperature to pellet large (>200 nt) RNAs. The supernatant was transferred to a new tube and small (<200 nt) RNAs were precipitated with pre-cooled isopropanol at −70 °C. Small RNAs were pelleted at 20,000 × g at 4 °C for 30 min, washed once with 70% pre-cooled ethanol, and dissolved with nuclease-free water. 10 μg of small RNA was fractionated on a 15% PAGE/7 M urea gel, and RNA from 17 nt to 26 nt was excised from the gel. RNA was extracted by soaking the gel in two gel volumes of NaCl TE buffer (0.3 M NaCl, 10 mM Tris-HCl, 1 mM EDTA pH 7.5) overnight. The supernatant was collected through a gel filtration column. RNA was precipitated with isopropanol, washed once with 70% ethanol, and resuspended with 15 μL nuclease-free water. RNA samples were treated with RppH to convert 22G-RNA 5' triphosphate to monophosphate in 1× reaction buffer, 10 U RppH (New England Biolabs), and 20 U SUPERase-In RNase Inhibitor (Invitrogen) for 3 h at 37 °C, followed by 5 min at 65 °C to inactivate RppH. RNA was then concentrated with the

RNA Clean and Concentrator-5 Kit (Zymo Research). Small RNA libraries were prepared according to the manufacturer's protocol of the NEBNext Multiplex Small RNA Sample Prep Set for Illumina-Library Preparation (New England Biolabs). NEBNext Multiplex Oligos for Illumina Index Primers were used for library preparation (New England Biolabs). Libraries were sequenced using an Illumina HiSeq4000 to obtain single-end 36 nt sequences at the University of Chicago Genomic Facility.

### RNA immunoprecipitation sequencing (RIP-seq)

A total of ~100,000 synchronized young adult animals were used for RIP-seq. Worm pellets were resuspended in equal volumes of immunoprecipitation buffer (20 mM Tris-HCl pH 7.5, 150 mM NaCl, 2.5 mM $MgCl_2$, 0.5% NP-40, 80 U/mL RNase Inhibitor (Thermo Fisher Scientific), 1 mM dithiothreitol, and protease inhibitor cocktail without EDTA (Promega)), and grinded in a glass grinder for 8–10 times. Lysates were clarified by spinning down at 15,000 rpm, 4 °C, for 15 min. Supernatants were incubated with the GFP-Trap magnetic agarose beads (ChromoTek) at 4 °C for 1 h. Beads were washed with wash buffer (20 mM Tris-HCl pH 7.5, 150 mM NaCl, 2.5 mM $MgCl_2$, 0.5% NP-40, and 1 mM dithiothreitol) six times, and then resuspended in TBS buffer for RNA extraction. Total RNA was extracted using the standard method with TRIzol reagent (Invitrogen). Small RNA libraries for RNA-seq were prepared as described above. Libraries were sequenced using an Illumina HiSeq4000 to obtain single-end 36-nt sequences at the University of Chicago Genomic Facility.

### Chemical crosslinking and co-immunoprecipitation of GLH-1

Chemical crosslinking of proteins was performed with DTME. ~100,000 synchronized flag::mCherry::GLH-1 and wild type (N2) young adults were collected and washed three times with M9. Two biological replicates for each genotype were used. M9 was discarded to the same amount of the worm volume, then DTME dissolved in Dimethyl Sulfoxide (DMSO) was added to a final concentration of 2 mM. Samples were incubated for 30 min at room temperature with occasional shaking before being washed three times with M9 to remove the crosslinker. Worm pellets were resuspended in equal volume of immunoprecipitation buffer (20 mM Tri-HCl pH 7.5, 150 mM NaCl, 2.5 mM MgCl2, 0.5% NP-40, 80 U/mL RNase Inhibitor (Thermo Fisher Scientific), 1 mM dithiothreitol, and protease inhibitor cocktail without EDTA (Promega)). Worm pellets were homogenized using glass homogenizer for 15–20 strokes on ice. Lysates were centrifuged at $14,000 \times g$ (Eppendorf Centrifuge 5424 R) for 10 min to remove insoluble material. Supernatants were incubated with 25 μL of Anti-Flag Magnetic Beads (bioLinkedin) for 2 h at 4 °C on an end-to-end rotator. The supernatant was removed and beads were washed with 1 mL of wash buffer (20 mM Tris-HCl pH 7.5, 150 mM NaCl, 2.5 mM $MgCl_2$, 0.5% NP-40) six times for 10 min each time, and with the final wash of 0.05% NP-40. Beads were incubated at 37 °C for 30 min in 50 μL decrosslinking buffer (50 mM Tris-HCl pH 7.5, 150 mM NaCl, 2 mM $MgCl_2$, 0.2% Tween-20, 10 mM dithiothreitol). The final samples were boiled in 2× SDS loading buffer at 100 °C for 5 min before mass spectrometry analysis.

### Mass spectrometry analysis

**Mass spectrometry.** The samples were subjected to SDS PAGE gel electrophoresis experiments (120 V, 1 hr) and visualized by Coomassie staining to obtain gel strip samples. Whole lanes were collected as single samples. 800 μL of 0.1 M NH4HCO3/30% acetonitrile (ACN) was added to samples, decolorized, and washed until the protein blue disappeared, and supernatant was removed. 800 μL of H2O was added to samples immediately to terminate reactions for 10 min, and the supernatant was removed. 800 μL 100 mM NH4HCO3 was added and samples were allowed to stand for 20 min, and the supernatant was removed. 40 μL 100 mM DTT and 360 μL 100 mM NH4HCO3 were

added to samples and samples were incubated at 56 °C for 30 min to reduce the protein; the supernatant was removed, 100 μL 100% ACN was added and incubated for 5 min, then aspirated. 280 μL 100 mM NH4HCO3 and 120 μL 200 mM 3-indole acetic acid (IAA, freshly prepared, stored in the dark) was added at room temperature and incubated for 20 min in the dark; the supernatant was removed,100 μL 100 mM NH4HCO3 was added, and left to stand at room temperature for 15 min to remove the supernatant; 100 μL 100% CAN was added and incubated for 5 min, aspirated and freeze-dried. 600 μL of 20 ng/μL Trypsin (Promega, V5113) was added to samples and samples were placed in a refrigerator at 4 °C for ~30 min to inflate the gel block; then 100 μL of 50 mM NH4HCO3 buffer was added and digested overnight at 37 °C. The enzymatic hydrolysate was aspirated and transferred to a new centrifuge tube, 100 μL of 60% ACN/0.1%TFA (trifluoroacetic acid) was added to the gel block samples and sonicated for 15 min, solution was aspirated and added to the previous solution. This extraction was repeated three times, combined, and lyophilized. After enzymolysis, the peptides were desalted using a C18 StageTip column, concentrated, and dried. Then the peptides were reconstituted with 0.1% formic acid aqueous solution for subsequent LC-MS/MS analysis

**LC-MS/MS analysis.** An appropriate amount of peptides from the sample was taken and used on the nanoliter flow rate Easy-nLC 1200 chromatography system (Thermo Scientific) for chromatographic separation. Buffer A was a 0.1% formic acid aqueous solution, and buffer B was a mixed solution of 0.1% formic acid, ACN, and water (where ACN was 80%). The column was equilibrated with 100% buffer A. Samples were injected into the Trap Column (100 μm × 20 mm, 5 μm, C18, Dr. Maisch GmbH) and then passed through the chromatographic analysis column (75 μm × 150 mm, 3 μm, C18, Dr. Maisch GmbH) for gradient separation at a flow rate of 300nL/min. The related liquid phase gradient was as follows: 0–3 min, linear gradient of liquid B from 2% to 8%; 3–43 min, linear gradient of liquid B from 8% to 28%; 43–51 min, linear gradient of liquid B from 28% to 40%; 51–52 min, linear gradient of liquid B from 40% to 100%; 52–60 min, linear gradient of liquid B maintained at 100%. After separation of the peptides, a Q Exactive HF-X mass spectrometer (Thermo Scientific) was used for DDA (Data Dependent Acquisition) mass spectrometry analysis. The analysis time is 60 min, the detection mode: positive ion, the precursor ion scan range: 350–1800 m/z, the resolution of the primary mass spectrometer is 60000 @m/z 200, AGC target 3e6, and the primary maximum IT: 50 ms. Peptide secondary mass spectrometry analysis is collected according to the following method: after each full scan (full scan), it is triggered to collect the secondary mass spectrum (MS2 scan) of the 20 highest intensity precursor ions, and the resolution of the secondary mass spectrum is 15000@m/z 200, AGC target: 1e5, secondary Maximum IT 25 ms MS2 Activation Type: HCD, Isolation window: 1.6 m/z, Normalized collision energy: 28. The mass spectrometry database search software MaxQuant 1.6.1.0 was used with the following protein database from Uniprot Protein Database: species *C. elegans* uniprot-*C. elegans* [6239]-27419-20210222.fasta. A PSM FDR < 0.01 and a protein FDR < 0.01 were used to assign positive protein identifications.

### RNA smFISH and PRG-1 immunohistochemistry

**smFISH of nos-3, gfp, and ceh-49 mRNA.** RNA smFISH was performed on dissected adult *C.elegans* gonads. For particular genotypes and conditions, experiments were performed with at least two technical replicates. Staged young adult worms were washed with M9 and resuspended in Dissection Buffer (PBS, 1 mM EDTA) then deposited onto 18 mm circular coverslips that were coated with 0.1% Poly-L-lysine solution. Gonads were dissected from whole worms using 25 G × 5/8 hypodermic needles directly onto coverslips in a 12-well tissue culture plate. All subsequent steps were performed directly in the 12-well plate. Gonads were fixed in 4% formaldehyde for 30 min at room

temperature. Gonads were then washed with PBS and dehydrated with 70% ethanol at 4 °C for one or two overnights. Gonads were rehydrated with PBS and washed once with FISH wash buffer (2× SSC, 10% formamide) at 37 °C. FISH probes were suspended in Hybridization Buffer (10% formamide, 2 mM vanadyl ribonucleoside complex, 20 mg/mL BSA, 10 mg/mL dextran sulfate, 2 mg/mL E.coli tRNA) 1:50. smFISH probes were created by Biosearch Technologies to target gfp mRNA, nos-3 mRNA, or ceh-49 mRNA. Oligo sequences can be found in Supplementary Data 2. gfp mRNA probes were conjugated to Quasar670 (Custom probe, Biosearch Technologies Cat. No. SMF-1065-5), nos-3 to CalFluor Red 610 (Custom probe, Biosearch Technologies Cat. No. SMF-1082-5), and ceh-49 to Quasar670 (Custom probe, Biosearch Technologies Cat. No. SMF-1065-5). Gonads were hybridized overnight at 37 °C in 12 well tissue plate wrapped tightly in parafilm. Gonads were washed with FISH wash buffer, stained with DAPI, and mounted onto 25 mm × 75 mm × 1 mm plain glass slides with ProLong Diamond Antifade Mountant. Samples were imaged on a Zeiss LSM800 Confocal Microscope at ×40 magnification using a Plan-Apochromat ×40/1.4 Oil Objective.

**smFISH of nos-3, gfp, and ceh-49 mRNA and antibody staining of PRG-1.** RNA smFISH and antibody staining were performed on dissected adult C.elegans gonads. For particular genotypes and conditions, experiments were performed with at least two technical replicates. Staged young adult worms were washed with M9 and resuspended in Dissection Buffer (PBS, 1 mM EDTA) then deposited into a watch glass. Adults were dissected using 25 G × 5/8 hypodermic needles and then transferred to 1.5 mL microcentrifuge tubes. All subsequent steps were performed in individual microcentrifuge tubes. Gonads were fixed in ice-cold methanol for 5 min, freeze cracked in liquid nitrogen for one minute, and submerged in ice-cold acetone for 5 min. Gonads were washed once with Antibody Wash Buffer (PBS, 1 mM EDTA, 0,1% Tween-20) and 50 µL of anti-PRG-1[4] suspended 1:200 in Antibody Suspension Buffer (PBS, 1 mM EDTA, 0.1% Tween-20, 20 mg/mL, 2 mM vanadyl ribonucleoside complex) was added. The anti-PRG-1 antibody is a custom-made anti-rabbit antibody provided by Dr. Craig Mello lab[4]. Tubes were shaken at 850 rpm overnight at 4 °C. Gonads were washed twice with Antibody Wash Buffer and 50 µL of anti-Rabbit Alexa488 (Jackson Labs Cat. No. 711-547-003) suspended 1:400 in Antibody Suspension Buffer was added. Tubes were shaken at 850 rpm for 2 h at room temperature. Gonads were washed once with Antibody Wash Buffer and once with 2× SSC. Samples were suspended in FISH Wash Buffer for 10 min at 37 °C. smFISH protocol, mounting, and imaging were then performed as above.

**Image processing and analysis.** Image processing and quantification of fluorescent puncta were performed using ImageJ. Single slice images of gonads were used for quantification. ROIs were selected, and areas of ROIs were measured. For smFISH and PRG-1 channels (Alexa488, CalFluor Red 610, and Quasar670), a Difference of Gaussian filter was applied to full image stacks with $\sigma_1 = 4$ and $\sigma_2 = 1$. Image thresholds were set manually and uniformly across all samples. Colocalization analysis was performed by measuring the total number of pixels in a stack which showed colocalization as a fraction of the total number of pixels in a stack pertaining to the molecule being measured. For example, colocalization of gfp mRNA with PRG-1 protein was measured by dividing the number of thresholded pixels in the gfp channel which overlap with thresholded pixels in the PRG-1 channel by the total number of thresholded pixels in the gfp channel. Images from 7–30 gonads were collected and quantified. Data were compared by student's t test.

### Sequencing data analysis

**RNA-seq.** Fastq reads were trimmed of adaptors using cutadapt[52]. Trimmed reads were aligned to the C.elegans genome build WS230 using bowtie2 ver 2.3.0[53]. After alignment, reads were overlapped with genomic features (protein-coding genes, pseudogenes, transposons) using bedtools intersect[54]. Reads per kilobase million (RPKM) values were then calculated for each individual feature by summing the total reads mapping to that feature, multiplied by 1e6, and divided by the product of the kilobase length of the feature and the total number of reads mapping to protein-coding genes. Protein-coding genes were used to normalize by sequencing depth because mRNA libraries were prepared by polyA tail selection, so reads mapping to features devoid of polyA tails are likely contaminants. RPKM values were then used in all downstream analyses using custom R scripts, which rely on packages ggplot2[55], reshape2[56], ggpubr[57], dplyr[58].

**sRNA-seq.** Fastq reads were trimmed using custom perl scripts. Trimmed reads were aligned to the C.elegans genome build WS230 using bowtie ver 1.2.1.1[59] with options -v 0 -best -strata. After alignment, reads that were between 17–40 nucleotides in length were overlapped with genomic features (rRNAs, tRNAs, snoRNAs, miRNAs, piRNAs, protein-coding genes, pseudogenes, transposons) using bedtools intersect[54]. Sense and antisense read mapping to individual miRNAs, piRNAs, protein-coding genes, pseudogenes, RNA/DNA transposons, simple repeats, and satellites were totaled and normalized to reads per million (RPM) by multiplying be 1e6 and dividing read counts by total mapped reads, minus reads mapping to structural RNAs (rRNAs, tRNAs, snoRNAs) because this sense reads likely represent degraded products. Reads mapping to multiple loci were penalized by dividing the read count by the number of loci they perfectly aligned to. Reads mapping to miRNAs and piRNAs were only considered if they matched the sense annotation without any overlap. In other words, piRNA and miRNA reads that contained overhangs were not considered mature piRNAs or miRNAs respectively. 22G-RNAs were defined as 21 to 23 nucleotide long reads with a 5′G that aligned to protein-coding genes, pseudogenes, or transposons. RPM values were then used in all downstream analyses using custom R scripts using R version 4.0.0[60], which rely on packages ggplot2[55], reshape2[56], ggpubr[57], dplyr[58]. To determine GLH-dependent 22G-RNA lists and PRG-1-dependent 22G-RNA lists, a Bayesian approach as described previously[61] was used: two models were compared to determine p-values. The first model states that a given gene has the same probability of accumulating positive reads between two samples. The second model states that a given gene's probability of accumulating positive reads is unique between two samples. The probability of each model for each gene was calculated and compared to yield the final probability (p value) for each gene's read accumulation being the same between the two samples. Significance was determined with Bonferroni correction $P < 0.05$.

**Metagene analysis.** Metagene profiles across gene lengths were calculated by computing the depth at each genomic position using 21–23 nucleotide long small RNA reads with a 5′G using bedtools genomecov[54]. A custom R script was then used to divide genes into 100 bins and sum the normalized depth within each bin. Groups of genes were then plotted using the sum of the normalize depth at each bin. To determine transcripts with particular CSR-1 IP enrichment at their 3′ ends, the proportion of 21–23 nucleotide long small RNA reads with a 5′ G in averaged wild-type CSR-1 IP libraries mapping to the last 15% of each transcript's length was compared to the proportion of reads mapping to the remaining 85% of each transcript's length. Non-enriched transcripts were accordingly defined as those with <15% of the reads mapping to the full-length transcript falling into that last 15% bin. Medium enrichment were those transcripts with 16% to 30% of their reads mapping to the last 15% of their mRNA length, and high enrichment were those transcripts with >30% of their reads mapping to the last 15% of their mRNA length. Metagene profiles relative to

piRNA targeting sites were calculated as the mean normalized reads per million at each nucleotide position using the indicated 22G-RNA reads. piRNA targeting positions were determined using the stringent piRNA targeting rules for the indicated group of transcripts according to the previously published piRNA targeting rules[10].

**HRDE-1/CSR-1 ratio.** Normalized reads (RPM) were calculated as described above. RPM values were compared between HRDE-1 IP and CSR-1 IP libraries and ratio levels were assigned as follows: highly CSR-1 favored transcripts were those CSR-1 targeted transcripts with a $\log_2$ (HRDE-1 IP/CSR-1 IP) of <−2.8, CSR-1 favored were those with a $\log_2$ of between −2.8 and −1.82, slightly CSR-1 favored were those with a $\log_2$ of between −1.82 and 0, and HRDE-1 favored were those CSR-1 targeted transcripts with a $\log_2$ >0.

### Reporting summary

Further information on research design is available in the Nature Research Reporting Summary linked to this article.

## Data availability

The data that support this study are available from the corresponding authors upon reasonable request. All sequencing data generated in this study (mRNA-seq, small RNA-seq, and RIP-seq) are available at the NCBI SRA database with accession number PRJNA802581. Small RNA-seq from the *parn-1* mutant[26] can be found with accession number SRS1021265. The MS proteomics data have been deposited to the ProteomeXchange Consortium via the PRIDE partner repository with the accession number PXD033506. Source data are provided with this paper.

## Code availability

All custom scripts are available upon request to the corresponding authors.

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

## Acknowledgements

We thank Dr. Edwin Ferguson, Dr. Karen Bennett, and members of the Lee lab for critical comments on the manuscript. Some strains used in this study were provided by the CGC, which is funded by NIH Office of Research Infrastructure Programs (P40 OD010440). This work is supported in part by NIH predoctoral training grant T32 GM07197 to J.B.; the National Center for Advancing Translational Sciences (NCATS) of the National Institutes of Health (NIH) grant 1UL1TR002389-01 to the Institute for Translational Medicine (ITM); the National Natural Science Foundation of China (grants 31771500 and 31922019) and the program for HUST Academic Frontier Youth Team (grant 2018QYTD11) to DZ; the Ministry of Science of Technology of Taiwan (MOST 108-2628-E-006-004-MY3 and MOST 110-2221-E-006-198-MY3 grants to W.-S.W. the NIH P01 grant (HD078253) to Z.W, the NIH grant R01-GM132457 to H.-C.L.

## Author contributions

H.-C.L. and D.Z. identified, supervised, and developed the core questions addressed in the manuscript. W.C. and J.S.B. performed most of the experiments and analyzed the results. T.H. performed the IP-MS analyses of GLH-1 complex. J.S.B. performed the bioinformatic analysis along with W.-S.W., S.Tu., and Z, W. H.-C.L. and J.S.B wrote the paper with contributions from all authors.

## Competing interests

The authors declare no competing interests.
