## [Peer Review File · Nature Communications]

Title: GLH/VASA helicases promote germ granule formation to ensure the fidelity of piRNA-mediated transcriptome surveillanceREVIEWER COMMENTS

Reviewer #1 (Remarks to the Author):

The manuscript by Chen et al. describes the role of the two GLH/VASA helicases in germ granule formation and their impact on small RNA biosynthesis and function. They have evaluated several mutants of germ granule proteins and their effects on germ granule formations in adults and embryos. The removal of GLH-1/4 is sufficient to disrupt the organization of the three granule condensates (P and Z granules and mutator foci). Surprisingly, even though PRG-1 is destabilized by mutation of GLH-1/4, the piRNAs accumulation is only mildly perturbed. Instead, they observed more severe effects in the 22G-RNAs generated in mutator foci. They evaluated the loading of these 22G-RNAs in the nuclear Argonaute HRDE-1 upon mutation of GLH-1/4. They identified some reduced 22G-RNAs from mRNAs targeting piRNAs with a consequently mild increase in the corresponding mRNA accumulation. They have also identified the novel acquisition of 22G-RNAs from genes usually protected by the Argonaute CSR-1 (a small proportion though, 6%). They have observed a corresponding decrease in the accumulation of these mRNAs.

Interestingly, the CSR-1 protected genes targeted by HRDE-1 in GLH-1/4 already have some level of 22G-RNA loading in HRDE-1 compared to all the other CSR-1 targets. Therefore, the authors speculate that this might be the reason for the misrouting 22G-RNAs into HRDE-1 from CSR-1 22G-RNAs. In general, most of the experiments are well executed with proper controls, supporting the paper's conclusion. The findings presented are a bit descriptive, and no mechanistic insights are provided to elucidate how GLH-1/4 affects granule formations and small RNA biogenesis. Moreover, some of their findings corroborate previously published results, which should be acknowledged in the text.

Main comments:

1. The authors should acknowledge the mass spec analysis of GLH-1 performed in Marnik et al. 2019. Here they have performed a more quantitative analysis (by LC-MS/MS) of the interactome of GLH-1 than the one presented by the authors in this manuscript and a previous manuscript (Chen et al. 2020). Most of the factors identified in this manuscript were also identified by Marnik et al. in a native condition of GLH-1 IPs. In addition, quantitative proteomic analysis of PRG-1 and CSR-1 complexes by Barucci et al. in a native condition also identified GLH-1 among the interacting proteins (especially in PRG-1 IPs where this interaction is quite strong and significant). Barucci et al. also identified other granule components in their PRG-1 proteomic such as DEPS-1, WAGO-1, CSR-1, and Z granule factor WAGO-4. Therefore, the proteomic analysis and results presented in this manuscript are not novel and should be noted in the text.

2. On page 6, the authors wrote, "Surprisingly, the localization of MUT-16 was significantly disrupted in glh-1 glh-4 double mutants (Figure 1D and S1C)". I think it is worth mentioning that Singh et al. 2021 previously showed that RNAi of four P granule proteins (GLH-1, GLH-4, PGL-1, and PGL3) destabilize the formation of Z granule and the mutator foci and consequently the biogenesis of PRG-1-dependent 22G-RNAs. Similarly, on page 14, the authors wrote, "22G-RNAs antisense to CSR-1 target genes remain

mostly unchanged in all mutants compared to wild type (Figure 5A, left and Figure S5A)". Singh et al. have shown that CSR-1 22G-RNAs are produced in the cytosol, and removing the four P-granule proteins by RNAi does not change the abundance of CSR-1 22G-RNAs. The authors should mention these previous findings in the text, corroborating their results.

3. On page 7, the authors wrote, "In the *glh-1* DQAD mutant, large PRG-1 and WAGO-4 aggregates are found in the cytoplasm with a significant reduction in perinuclear PRG-1 and WAGO-4 foci (Figure 2B). In addition, these abnormal, cytoplasmic aggregates are not properly sorted to the germ cell lineage, leading to the presence of these foci in somatic lineages (Figure 2B)". The characterization of the DQAD mutant and its effect on cytoplasmic granule aggregate in the germline and embryos has been previously shown by Marnik et al. 2019. They have also analyzed the impact of DQAD mutation on PGL-1 and PRG-1 localization. Therefore, the novel result here is the addition of WAGO-4 localization, which is not surprising given that proteomics data from the DQAD mutant protein in Marnik et al. also showed increased interaction with Argonaute proteins, such as CSR-1, PRG-1, WAGO-1, and C04F12.1. Moreover, previous results from the Lee lab also indicate the presence of large cytoplasmic aggregate in *glh-1* DQAD mutant adults and embryos, including GLH-1, PRG-1, CSR-1, and PGL-1 (Chen et al. 2020). Therefore, the results presented in this current manuscript should be considered in light of these previous publications.

4. In figure 4C, the authors should explain why in *meg-3/4* mutant, the mRNA of "reduced HRDE-1 targeting" and "other WAGO genes" are globally upregulated even though they have a mild reduction in corresponding 22G-RNAs. Also, maybe it would be better to remove data on DEPS-1 and PGL-1 since the authors do not show a complete dataset for these mutants (mRNAs and 22G-RNAs). The same comment is in Figure 5C.

5. In figure 2E, the authors show examples of smFISH aiming to demonstrate that the absence of GLH-1/4 affects the retention of the perinuclear smFISH signal compared to cytoplasmic smFISH detection. The authors should provide a quantification of these effects. Also, they might need to overlap the signal of smFISH with P-granule localization to make sure those perinuclear signals are mRNA molecules retained in germ granules.

Reviewer #2 (Remarks to the Author):

Chen et al. in the study entitled 'GLH/VASA helicases promote germ granule formation to ensure the fidelity of piRNA-mediated transcriptome surveillance' performed an study to investigate if GLH/VASA helicase mutants have defects in forming perinuclear condensates containing PIWI and other small RNA cofactors. This investigation is performed *Caenorhabditis elegans* worm, and expression of RNA was investigated. In parallel, a wide microscopy study was performed and mass spectrometry was used to investigate co-precipitated proteins. Regarding this latest technology, I've been contacted to review the proteomics part of this manuscript according to my expertise.

The technology used and the mass spectrometry section of materials and methods is correct. LC-MSMS part is very detailed. But in my opinion, there are aspects that can be further improved, as for example the details about the in-gel digestion protocol. Additionally, a data analysis results section regarding how authors processed the proteomics data is totally missing. The number of replicates in the text is not stated. I realized that in the table contained in Figure 1 authors shows number of spectra detected in two replicates. This is also an important concern, regarding the validity of this data.

Here I include a point-by-point revision of the proteomics-related aspects:

Lines 309-311. When authors mention the immunoprecipitation study it is better to talk about proteins than genes.

Section Mass spectrometry analysis. More details about in-gel digestion are needed. Otherwise, cite the reference used to follow the protocol of in-gel digestion if this protocol is already published in another article. Information regarding amount of trypsin used, if all the band was collected as a single sample for subsequent desalting, or gel was cut in "fractions" first, etc.

A section about data analysis for mass spectrometry is totally missing. It is not clear to me how authors can choose this such a small number of proteins of the immunoprecipitation study. Did authors obtained only 6 co-precipitated proteins together with glh-1? If more proteins were identified please specify how the selected proteins were chosen.

Datasets with proteomics identifications are missing in supplementary.

How many biological and technical replicates were performed in this proteomics study?

What kind of trypsin did authors use? Sequencing-grade?

Line 549. Remove "at"

Line 551. Rewrite the sentence in past tense.

Figure 1A. Did authors consider that two replicates are enough to obtain statistical significance in the proteomics study?

Reviewer #3 (Remarks to the Author):

22G RNAs are effector sRNAs whose structural properties appear identical but can promote very different outcomes: gene silencing or gene expression. A central puzzle in this regard concerns how sRNAs choose to or are chosen to associate with the CSR-1 anti-silencing Argonaute protein or the HRDE-1 pro-silencing Argonaute protein. Overall, this manuscript addresses this problem by showing that P granule structure may promote proper sorting of 22G small RNAs to CSR-1 via a population of target mRNAs that reside in P granules. Release of CSR-1-targeted mRNAs from P granules allows Piwi-associated piRNAs to create pro-silencing 22G RNAs that then associate with HRDE-1. One irony of these findings is that Piwi itself is concentrated in P granules and this association is profoundly disrupted by GLH Vasa dysfunction. Piwi may therefore be capable of promoting 22G RNA biogenesis at a location that is distinct from P granules, which is an unexpected discovery.

Chen and colleagues address a role for the germ granule protein Vasa in creating a perinuclear environment whose architecture is important for the association of small RNAs with the anti-silencing Argonaute CSR-1. In the absence of Vasa orthologues GLH-1 and GLH-4, a subset of genes that are

protected from silencing by CSR-1-associated 22G RNAs become silenced. The mRNAs for some of these genes normally localize to P granules where they presumably associate with CSR-1 and may guide these mRNAs to mutator bodies where 22G RNAs are made that are transported back into P granules to associate with CSR-1 and then migrate in some fashion back into the nucleus to promote gene expression. In the absence of GLH/Vasa, a subset of CSR-1 target genes that normally display CSR-1-associated 22G RNAs across their gene bodies display reduced levels of transcription, and are targeted by HRDE-1-associated 22G RNAs that promote nuclear silencing. The CSR-1 targets that become silenced in the absence of P granule proteins are in a 'mixed' category of genes that have 22G RNAs that associate with both HRDE-1 and CSR-1 in wildtype animals. Therefore, the category of CSR-1 targets that becomes silenced upon P granule dysfunction may be 'partially silenced' yet at the same time remains protected by CSR from silencing. A distinct category of CSR-1 targets remains protected from silencing by HRDE-1 and is enriched for 22G RNAs that map to the 3' UTRs of these genes. CSR-1 targets that remain protected by CSR-1 in the absence of GLH/Vasa might be consistent with weak but significant perinuclear localization of CSR-1.

Overall, this interesting manuscript reveals insight into how pro- and anti-silencing sRNAs targeting is shaped. Because the anti-silencing function of sRNAs remains a mysterious problem that is best understood in *C. elegans*, this manuscript offers insight that would be difficult or impossible to learn in another experimental system. The authors provide insight into how pro- and anti-silencing pathways are wired, and their unexpected discoveries represent a lot of work. A discussion of distinctions between genes whose expression is and is not affected by P granule function could more clearly convey what the authors have learned and what remains to be understood regarding how 22G RNAs that are structurally indistinguishable impart opposing effects at target loci within nuclei.

Comments:

1. Please move the model to a main figure, as this is essential for understanding the author's conclusions.
2. A central conclusion is that CSR-1 function is more strongly tied to P granule structure for a subset of CSR-1 targets that is misregulated and normally has some HRDE-1-associated 22G RNAs. If 717 CSR-1 targets exhibited increased 22G RNA levels and 320 display HRDE-1-associated 22G RNA levels, then what about the other 400 CSR-1 targets? Do these genes display increased mRNA levels in *glh* mutants? Are these 22G RNAs associated with CSR-1 such that their numbers are increased in CSR-1 IP's? One reason that this might be important is that a steady-state level of 22G RNAs may be funneled to CSR-1, so if 320 genes have their 22G RNAs shifted to HRDE-1, then perhaps there is a corresponding increase in CSR-1-associated 22G RNAs for other target genes. These could be from natural CSR-1 targets or from Piwi targets that are no longer silenced.
3. Discussion: 'piRNA silencing of non-self is reduced. Simultaneously, hundreds of self RNAs are silenced'. It would be helpful if the authors could summarize in the Discussion how many CSR-1 targets do not change their expression when GLH is mutant relative to the fraction that is affected. This might convey a larger picture understanding how important Vasa or P granules are for CSR-1 function. If the

majority of piRNA and CSR-1 targets remain unaltered, what does this mean about how P granules shape gene expression? What can be said about the rest of the targets? This discussion might allow the magnitude of the pro- and anti-silencing defects to be understood.

4. There is a model from the Ketting group that pro- and anti-silencing pathways become misregulated in the context of prg-1 sterility when 22G RNAs are restored. I did not notice this reference or a comparison of the effects observed by Ketting and colleagues with the authors' observations.

5. What fraction of genes whose CSR-1-associated 22G RNAs are spread out along gene bodies remain expressed? If this is small, then this may be the sole decisive characteristic of CSR-1 targets that become silenced when GLH is mutant.

6. For Piwi targets that become expressed, perhaps acknowledge that these may differ from those that remain silenced, and in the Discussion explicitly state that understanding this distinction remains an important question in the field. An alternative might be to offer a more wholistic model for how CSR and Piwi misregulation is coordinately achieved.

7. The authors have made good progress by defining populations of RNA targets whose expression is modulated in response to P granule perturbation. If P granule disruption fails to affect expression of most Piwi and CSR-1 targets, perhaps there are small RNA amplification loops that occur in the nucleus or cytoplasm or in a residual P granule structure or in Z or Mutator granules. Even if one cannot detect the presence of a factor near the nucleus by microscopy, this factor could be there in small amounts. An open question may be precisely how the P granule disruption that the authors report affects the structure and function of P granules and associated bodies. Perhaps acknowledge this caveat in the Discussion.

8. The authors report that mutator foci are affected by GLH / Vasa but this is missing from the discussion. Mutator foci may be where 22G RNA biogenesis occurs, so perhaps it is alterations to these foci that are responsible for some alterations to 22G RNA populations. Do the authors feel that their RNA FISH clearly distinguishes localization to P granules rather than to Mutator foci? If so, why is the Discussion mostly focused on a role for P granules in their observations?

9. How does mRNA localization to P granules lead to 22G RNA biogenesis in mutator bodies that funnels 22G RNAs back to the P granule and to the correct Argonaute protein? This may not be well understood but is probably relevant to data in this paper. Do the authors imagine that CSR-1 associates with CSR-1 mRNA targets in P granules and that the 22G RNAs made from rare mRNAs in the mutator bodies then get funneled back into the P granule where the CSR-1-associated mRNAs soak up local concentrations of 22G RNAs into a CSR-1 sub-domain that might be critical for the 22G sorting process? If so, how does this tie into the altered 22G and mRNA expression observed for a sub-set of Piwi and CSR targets? What is known about mRNA localization to Mutator bodies where RDRPs are concentrated? Some of these points might be offered in the Discussion to create a more coherent understanding of the framework of the problem being studied.

10. page 9. These results indicate that localization of piRNA factors at perinuclear and cytoplasmic P granules can both contribute to their function in piRNA silencing. How do the authors data clearly show that cytoplasmic P granules promote silencing? Perhaps soften this conclusion?

11. page 12. If glh-1 single mutants are more compromised for disrupted 22G and mRNA levels, perhaps the authors should ask what characteristics are different or shared between glh-1 single mutants and glh-1 glh-4 mutants.

Minor:

1. 'In *C. elegans*, piRNAs and other small RNA pathways factors' – pathway factors

2. Page 6. Point out that the number of CSR-1 foci is similar but that the overall level of perinuclear CSR-1 is reduced.

3. line 317. In other mutants defecting in cytoplasmic and/or. defective

4. line 346. 'an aberrantly silenced CSR-1 mRNA, and noticed that ceh-49 mRNAs were expressed'. an aberrantly silenced mRNA from a gene that is normally protected from silencing by CSR-1.

5. 'These results are consistent with the model that the P granule localization of some mRNA transcripts is critical for their protection from piRNA silencing.' If Piwi is also in P granules why are there more pro-silencing sRNAs made?

6. line 362. 'they are no longer distinct from WAGO genes'. that they no longer resemble WAGO targets?

Point-by-point response to the reviewers' comments

Reviewer #1 (Remarks to the Author):

The manuscript by Chen et al. describes the role of the two GLH/VASA helicases in germ granule formation and their impact on small RNA biosynthesis and function. They have evaluated several mutants of germ granule proteins and their effects on germ granule formations in adults and embryos. The removal of GLH-1/4 is sufficient to disrupt the organization of the three granule condensates (P and Z granules and mutator foci). Surprisingly, even though PRG-1 is destabilized by mutation of GLH-1/4, the piRNAs accumulation is only mildly perturbed. Instead, they observed more severe effects in the 22G-RNAs generated in mutator foci. They evaluated the loading of these 22G-RNAs in the nuclear Argonaute HRDE-1 upon mutation of GLH-1/4. They identified some reduced 22G-RNAs from mRNAs targeting piRNAs with a consequently mild increase in the corresponding mRNA accumulation. They have also identified the novel acquisition of 22G-RNAs from genes usually protected by the Argonaute CSR-1 (a small proportion though, 6%). They have observed a corresponding decrease in the accumulation of these mRNAs.

Interestingly, the CSR-1 protected genes targeted by HRDE-1 in GLH-1/4 already have some level of 22G-RNA loading in HRDE-1 compared to all the other CSR-1 targets. Therefore, the authors speculate that this might be the reason for the misrouting 22G-RNAs into HRDE-1 from CSR-1 22G-RNAs. In general, most of the experiments are well executed with proper controls, supporting the paper's conclusion. The findings presented are a bit descriptive, and no mechanistic insights are provided to elucidate how GLH-1/4 affects granule formations and small RNA biogenesis. Moreover, some of their findings corroborate previously published results, which should be acknowledged in the text.

We agree that it is interesting that GLH-1 is required for preventing mis-silencing of a subset of CSR-1 genes that seems to be silencing prone (targeted by both CSR-1 and surprisingly a bit by HRDE-1). We thank the reviewer for their insightful comments and their careful evaluation of our work. We also thank for the reviewer for pointing out previous work related to this study and we have now properly acknowledged their findings.

Main comments:

1. The authors should acknowledge the mass spec analysis of GLH-1 performed in Marnik et al. 2019. Here they have performed a more quantitative analysis (by LC-MS/MS) of the interactome of GLH-1 than the one presented by the authors in this manuscript and a previous manuscript (Chen et al. 2020). Most of the factors identified in this manuscript were also identified by Marnik et al. in a native condition of GLH-1 IPs. In addition, quantitative proteomic analysis of PRG-1 and CSR-1 complexes by Barucci et al. in a native condition also identified GLH-1 among the interacting proteins (especially in PRG-1 IPs where this interaction is quite strong and significant). Barucci

et al. also identified other granule components in their PRG-1 proteomic such as DEPS-1, WAGO-1, CSR-1, and Z granule factor WAGO-4. Therefore, the proteomic analysis and results presented in this manuscript are not novel and should be noted in the text.

We agree that the data presented in Marnik *et al* and in Barucci *et al* largely corroborate the evidence we present here from our own LC-MS/MS analysis, and we have now properly acknowledged their findings in our manuscript. While the experiment performed in Marnik *et al* using native GLH-1 IP is very similar to our LC-MS/MS experiment, here we additionally found evidence of interaction between GLH-1 and Z granule factor WAGO-4. The PRG-1 interactome reported in Barucci *et al* demonstrate an interaction between P, Mutator, and Z granule factors, and we have now acknowledged this important contribution in our manuscript.

2. On page 6, the authors wrote, “Surprisingly, the localization of MUT-16 was significantly disrupted in *glh-1 glh-4* double mutants (Figure 1D and S1C)”. I think it is worth mentioning that Singh et al. 2021 previously showed that RNAi of four P granule proteins (GLH-1, GLH-4, PGL-1, and PGL3) destabilize the formation of Z granule and the mutator foci and consequently the biogenesis of PRG-1-dependent 22G-RNAs. Similarly, on page 14, the authors wrote, “22G-RNAs antisense to CSR-1 target genes remain mostly unchanged in all mutants compared to wild type (Figure 5A, left and Figure S5A)”. Singh et al. have shown that CSR-1 22G-RNAs are produced in the cytosol, and removing the four P-granule proteins by RNAi does not change the abundance of CSR-1 22G-RNAs. The authors should mention these previous findings in the text, corroborating their results.

We thank the reviewer for pointing out this consistency with the previous report. We have now reported this connection and acknowledged the previous work in the text.

3. On page 7, the authors wrote, “In the *glh-1* DQAD mutant, large PRG-1 and WAGO-4 aggregates are found in the cytoplasm with a significant reduction in perinuclear PRG-1 and WAGO-4 foci (Figure 2B). In addition, these abnormal, cytoplasmic aggregates are not properly sorted to the germ cell lineage, leading to the presence of these foci in somatic lineages (Figure 2B)”. The characterization of the DQAD mutant and its effect on cytoplasmic granule aggregate in the germline and embryos has been previously shown by Marnik et al. 2019. They have also analyzed the impact of DQAD mutation on PGL-1 and PRG-1 localization. Therefore, the novel result here is the addition of WAGO-4 localization, which is not surprising given that proteomics data from the DQAD mutant protein in Marnik et al. also showed increased interaction with Argonaute proteins, such as CSR-1, PRG-1, WAGO-1, and C04F12.1. Moreover, previous results from the Lee lab also indicate the presence of large cytoplasmic aggregate in *glh-1* DQAD mutant adults and embryos, including GLH-1, PRG-1, CSR-1, and PGL-1 (Chen et al. 2020). Therefore, the results presented in this current manuscript should be considered in light of these previous publications.

We have now made mention of this previously published data as we agree our findings here are largely not novel. We agree that the WAGO-4 localization is not altogether

surprising given our own LC-MS/MS data showing WAGO-4 interaction with GLH-1.

4. In figure 4C, the authors should explain why in *meg-3/4* mutant, the mRNA of “reduced HRDE-1 targeting” and “other WAGO genes” are globally upregulated even though they have a mild reduction in corresponding 22G-RNAs. Also, maybe it would be better to remove data on DEPS-1 and PGL-1 since the authors do not show a complete dataset for these mutants (mRNAs and 22G-RNAs). The same comment is in Figure 5C.

We agree that the lack of a complete dataset for *deps-1* and *pgl-1* mutants makes these figures more difficult to interpret. We have removed the mRNA boxplots from these panels and moved mRNA expression data to the supplement. Additionally, we have expanded our analysis to include small RNA expression data from *mip-1/2* mutants, which have been shown to severely disrupt P granule integrity. We found that these mutants show very similar 22G-RNA defects, consistent with our model and granule disruption leads to subtle but specific mis-regulation.

The reviewer’s conclusion that all WAGO targeted mRNAs are globally upregulated despite a “mild” reduction in 22G-RNAs may have been due to our perhaps misleading representation of the data in Figure 4C. For the “Other WAGO genes” category, the median value for the log₂ 22G-RNA change in *meg-3/4* mutants was -1.09 compared to the log₂ mRNA change in *meg-3/4* mutants of 0.48. Therefore, the 22G-RNA changes were actually generally more extreme than the mRNA changes. This finding of profound 22G-RNA production defects in the *meg-3/4* mutant is consistent with previously published work from the Seydoux and Rechavi labs (Ouyang *et al* 2019 and Lev *et al* 2019, respectively). Because we needed to change the y-axis values in the 22G-RNA plots to accommodate the more extreme *glh-1* mutant datasets, a direct comparison between the mRNA and 22G-RNA boxplot panels was not possible. Due to our having now moved the mRNA expression panels to the supplement, we believe this confusion can now be avoided. Finally, the WAGO 22G-RNA changes in *meg-3/4* mutants is quite predictive of which mRNAs will show the most upregulation relative to wild type animals. 66% of WAGO targeted genes with reduced 22G-RNA expression show elevated mRNA expression:

5. In figure 2E, the authors show examples of smFISH aiming to demonstrate that the absence of GLH-1/4 affects the retention of the perinuclear smFISH signal compared to cytoplasmic smFISH detection. The authors should provide a quantification of these effects. Also, they might need to overlap the signal of smFISH with P-granule localization to make sure those perinuclear signals are mRNA molecules retained in germ granules.

We thank the reviewer for this suggestion. We were able to perform additional experiments which clarified our observation substantially. We tried two methods to quantify perinuclear localization, one based on the proximity of mRNA signal to DAPI (chromatin) signal and the other based on colocalization with P granule factor PRG-1. We found this second method to be more reliable.

In the first method, inferred from the DAPI staining where nuclei were positioned in 3D space, then measured the distance between mRNA foci and the inferred nuclear envelope. We found that this quantitation method was highly error-prone as we could not uniformly assign the perinuclear character of PRG-1, which is known to be highly enriched at the perinucleus. We think this difficulty comes from the uneven distribution of chromatin in germline nuclei, and the syncytial nature of the adult gonad which sometimes results in very crowded nuclei upon dissection and fixation. To accurately determine the true perinucleus, some nuclear membrane marker would be better suited and could be very useful in the future.

Simultaneously, we developed our smFISH protocol to be compatible with PRG-1 staining so that we could monitor P granule localization directly. We found that using our immunofluorescence-compatible smFISH protocol allowed us to measure the P granule distribution of mRNA signal directly using colocalization with PRG-1. To perform this analysis properly, we used a version of the piRNA reporter strain that does not contain the silencing piRNA. We needed this additional strain to measure the effect of silencing on P granule colocalization because PRG-1 is dispersed in the *glh-1 glh-4* mutant, so it was not suitable to answer this question on its own. We have added this extensive analysis as Figure S2C. We found that when *gfp* mRNA is silenced, there is a significant increase in colocalization with PRG-1 protein. This is not the case for the germline expressed gene *nos-3*. Although the increase in colocalization was significant, the majority of the *gfp* mRNA signal in the silenced reporter strain does not colocalize with PRG-1. This could be due to partial colocalization with P granule adjacent bodies like the Mutator or Z granule. Because we only observed this modest effect, we discussed this caveat and softened our assertion that silenced *gfp* localizes to the perinucleus. We used the same protocol to directly test our assertion for the CSR-1 target that becomes silenced in P granule mutants – *ceh-49*. Similarly, we found that *ceh-49* mRNA does indeed show significantly more P granule accumulation than the germline expressed mRNA *nos-3*, but again the effect was mild. We have added this analysis as Figure S5C.

Reviewer #2 (Remarks to the Author):

Chen et al. in the study entitled 'GLH/VASA helicases promote germ granule formation to ensure the fidelity of piRNA-mediated transcriptome surveillance' performed an study to investigate if GLH/VASA helicase mutants have defects in forming perinuclear condensates containing PIWI and other small RNA cofactors. This investigation is performed in *Caenorhabditis elegans* worm, and expression of RNA was investigated. In parallel, a wide microscopy study was performed and mass spectrometry was used to investigate co-precipitated proteins. Regarding this latest technology, I've been contacted to review the proteomics part of this manuscript according to my expertise. The technology used and the mass spectrometry section of materials and methods is correct. LC-MSMS part is very detailed. But in my opinion, there are aspects that can be further improved, as for example the details about the in-gel digestion protocol.

We thank the reviewer for their diligence in reviewing our mass spectrometry data. We have expanded our description of the mass spectrometry methods, and we have addressed the comments point by point below. We believe that this important data in our paper is now much more understandable and will be easier for others to reproduce or reference as a result.

Additionally, a data analysis results section regarding how authors processed the proteomics data is totally missing. The number of replicates in the text is not stated. I realized that in the table contained in Figure 1 authors shows number of spectra detected in two replicates. This is also an important concern, regarding the validity of this data.

We have added a greatly expanded section in the methods detailing exactly how the proteomics data was processed. In this manuscript, we have done 2 biological replicates in our Mass spectrometry experiments, and the detailed protein identification results are provided in the supplementary table. The Mascot scores are provided in the supplemental table to reflect the confidence in protein identification.

Here I include a point-by-point revision of the proteomics-related aspects: Lines 309-311. When authors mention the immunoprecipitation study it is better to talk about proteins than genes.

Here, we are referring to the small RNAs sequencing data from immunoprecipitated protein complexes. Because these Argonaute protein complexes use these bound small RNAs to target mRNAs, we used this terminology to refer to the small RNA-targeted genes which share sequence complementarity with the sequenced protein bound small RNAs.

Section Mass spectrometry analysis. More details about in-gel digestion are needed. Otherwise, cite the reference used to follow the protocol of in-gel digestion if this protocol is already published in another article. Information regarding amount of trypsin

used, if all the band was collected as a single sample for subsequent desalting, or gel was cut in “fractions” first, etc.

We have expanded our methods section to more explicitly detail the exact methodology used.

A section about data analysis for mass spectrometry is totally missing. It is not clear to me how authors can choose this such a small number of proteins of the immunoprecipitation study. Did authors obtained only 6 co-precipitated proteins together with glh-1? If more proteins were identified please specify how the selected proteins were chosen.

We obtained 231 and 242 co-precipitated proteins in GLH-1 IP replicates that passed the identification threshold of $FDR < 0.01$. This is compared to 30 and 11 co-precipitated proteins in the untagged control IP replicates. We selected the 6 proteins emphasized in Figure 1 because they are known factors of small RNA pathways. The full mass spectrometry dataset can be found in the supplemental table, which also contains Mascot scores to indicate statistical confidence in the correct protein identification.

Datasets with proteomics identifications are missing in supplementary.

We have supplied the full list of enriched protein identifications in the supplemental table. We hope they will be useful to other labs interested in GLH-1 and P granule complexes.

How many biological and technical replicates were performed in this proteomics study? What kind of trypsin did authors use? Sequencing-grade?

We thank the reviewer for their attention to these essential details. We have added these to the methods. Briefly, two biological replicates were performed, and sequencing grade trypsin was used (Promega V5113).

Line 549. Remove “at”

Line 551. Rewrite the sentence in past tense.

Figure 1A. Did authors consider that two replicates are enough to obtain statistical significance in the proteomics study?

While more replicates will further increase our confidence, we found both replicates showed a highly similar set of identified proteins, and the several proteins emphasized in our study were never detected in untagged control IP experiments. As a consequence, we can be confident that these proteins are present in the GLH-1 immunoprecipitated complex. In addition, the identification of these factors passed the stringent statistical requirement imposed using MaxQuant software (PSM and protein FDR set at 0.01)

Reviewer #3 (Remarks to the Author):

22G RNAs are effector sRNAs whose structural properties appear identical but can promote very different outcomes: gene silencing or gene expression. A central puzzle in this regard concerns how sRNAs choose to or are chosen to associate with the CSR-1 anti-silencing Argonaute protein or the HRDE-1 pro-silencing Argonaute protein. Overall, this manuscript addresses this problem by showing that P granule structure may promote proper sorting of 22G small RNAs to CSR-1 via a population of target mRNAs that reside in P granules. Release of CSR-1-targeted mRNAs from P granules allows Piwi-associated piRNAs to create pro-silencing 22G RNAs that then associate with HRDE-1. One irony of these findings is that Piwi itself is concentrated in P granules and this association is profoundly disrupted by GLH Vasa dysfunction. Piwi may therefore be capable of promoting 22G RNA biogenesis at a location that is distinct from P granules, which is an unexpected discovery.

Chen and colleagues address a role for the germ granule protein Vasa in creating a perinuclear environment whose architecture is important for the association of small RNAs with the anti-silencing Argonaute CSR-1. In the absence of Vasa orthologues GLH-1 and GLH-4, a subset of genes that are protected from silencing by CSR-1-associated 22G RNAs become silenced. The mRNAs for some of these genes normally localize to P granules where they presumably associate with CSR-1 and may guide these mRNAs to mutator bodies where 22G RNAs are made that are transported back into P granules to associate with CSR-1 and then migrate in some fashion back into the nucleus to promote gene expression. In the absence of GLH/Vasa, a subset of CSR-1 target genes that normally display CSR-1-associated 22G RNAs across their gene bodies display reduced levels of transcription, and are targeted by HRDE-1-associated 22G RNAs that promote nuclear silencing. The CSR-1 targets that become silenced in the absence

of P granule proteins are in a 'mixed' category of genes that have 22G RNAs that associate with both HRDE-1 and CSR-1 in wildtype animals. Therefore, the category of CSR-1 targets that becomes silenced upon P granule dysfunction may be 'partially silenced' yet at the same time remains protected by CSR from silencing. A distinct category of CSR-1 targets remains protected from silencing by HRDE-1 and is enriched for 22G RNAs that map to the 3' UTRs of these genes. CSR-1 targets that remain protected by CSR-1 in the absence of GLH/Vasa might be consistent with weak but significant perinuclear localization of CSR-1.

Overall, this interesting manuscript reveals insight into how pro- and anti-silencing sRNAs targeting is shaped. Because the anti-silencing function of sRNAs remains a mysterious problem that is best understood in *C. elegans*, this manuscript offers insight that would be difficult or impossible to learn in another experimental system. The authors provide insight into how pro- and anti-silencing pathways are wired, and their unexpected discoveries represent a lot of work. A discussion of distinctions between genes whose expression is and is not affected by P granule function could more clearly convey what the authors have learned and what remains to be understood regarding how 22G RNAs that are structurally indistinguishable impart opposing effects at target loci within nuclei.

We thank the reviewer for their assessment of our manuscript and their constructive critiques that have undoubtedly improved our work. We have addressed individual comments below.

Comments:

1. Please move the model to a main figure, as this is essential for understanding the author's conclusions.

We have moved the model from the supplement to Figure 6.

2. A central conclusion is that CSR-1 function is more strongly tied to P granule structure for a subset of CSR-1 targets that is misregulated and normally has some HRDE-1-associated 22G RNAs. If 717 CSR-1 targets exhibited increased 22G RNA levels and 320 display HRDE-1-associated 22G RNA levels, then what about the other 400 CSR-1 targets? Do these genes display increased mRNA levels in *glh* mutants? Are these 22G RNAs associated with CSR-1 such that their numbers are increased in CSR-1 IP's? One reason that this might be important is that a steady-state level of 22G RNAs may be funneled to CSR-1, so if 320 genes have their 22G RNAs shifted to HRDE-1, then perhaps there is a corresponding increase in CSR-1-associated 22G RNAs for other target genes. These could be from natural CSR-1 targets or from Piwi targets that are no longer silenced.

One major reason for the drop from 717 aberrantly silenced CSR-1 targets in the total small RNA sequencing experiment to 320 targets showing increased HRDE-1 associated 22G-RNAs is that different requirements were set up for obtaining these genes; the 717 genes came from a calculation that only required a two-fold difference of 22G abundance between the mutant and wild type. For HRDE-1 IP-associated 22G-RNAs, we not only applied the two-fold cutoff, but also added a statistical cutoff of $p < 0.05$. We applied an additional cutoff for HRDE-1 IP since we sought to use this stricter gene list to better clarify precisely which genes were most affected by *glh* loss.

To examine whether those targets which have elevated 22G accumulation in *glh-1/4* mutants but did not meet the criteria of elevated HRDE-1 IP 22G-RNAs in *glh-1* mutants may have a distinct balance between HRDE-1 and CSR-1 IP, we remade Figure 5G using this group of genes (see below). However, we see a similar trend in the IP ratio, where this group has higher levels of HRDE-1/CSR-1 than other CSR-1 genes in the wild type background and this ratio further increases for the more stringent list, suggesting that these 600 CSR-1 targets that do not meet the statistical cutoff, they can still be considered part of this group of distinct aberrantly silenced targets that rely on germ granules for protection from silencing:

3. Discussion: ‘piRNA silencing of non-self is reduced. Simultaneously, hundreds of self RNAs are silenced’. It would be helpful if the authors could summarize in the Discussion how many CSR-1 targets do not change their expression when GLH is mutant relative to the fraction that is affected. This might convey a larger picture understanding how important Vasa or P granules are for CSR-1 function. If the majority of piRNA and CSR-1 targets remain unaltered, what does this mean about how P granules shape gene expression? What can be said about the rest of the targets? This discussion might allow the magnitude of the pro- and anti-silencing defects to be understood.

We thank the reviewer for raising these interesting points that help us better present our findings about the role of VASA and P granules in gene regulation in a larger picture. A relevant discussion has been now added in the discussion section.

There are ~ 6% (or 15% with less conservative criteria) of CSR-1 targets exhibiting increased HRDE-1 22G RNAs in VASA mutants, and 16% (or 41% with less conservative criteria) of WAGO targets exhibiting reduced HRDE-1 22G RNAs in VASA mutants. These results demonstrate that a significant portion of germline transcripts are mis-regulated by HRDE-1 22G-RNAs in VASA mutants. However, as both WAGO and CSR-1 22G-RNAs can establish epigenetic memories (Shirayama et al, 2012 and Conine et al., respectively), the silenced or expressed state of many germline

transcripts may be preserved in the absence of P granules. In this model, those mRNAs which did not establish robust epigenetic memories would be those that exhibit more mis-regulation in P granule mutants. Indeed, we observed that PRG-1 dependent 22G-RNA targets, which depend on PRG-1/piRNAs at each generation to trigger gene silencing, are those which exhibit a greater reduction of 22G-RNAs on WAGO targets in *glh-1/4* mutants (Figure S4F).

In addition, a previous study from the Ketting lab has demonstrated that re-establishment of the 22G-RNA system in *prg-1* mutants leads to gene-mis-regulation in a stochastic manner that varies between worms⁴⁵. As our measurements of 22G-RNAs or mRNAs are from hundreds of thousands of worms, stochastic activation or silencing that may exist in individual worms may not be detected. Examining whether P granule mutant worms exhibit aberrant activation or silencing of germline transcripts stochastically will be interesting in future studies. Taken together, our current model is that P granules provide a critical environment that allows distinct Argonautes (including silencing PRG-1 and anti-silencing CSR-1 Argonautes) to survey their targets. Loss of P granules thus leads to a failure of Argonautes to properly identify their targets and mis-regulation of hundreds of germline targets results. Some other evidence that support this model are described in the responses below.

4. There is a model from the Ketting group that pro- and anti-silencing pathways become misregulated in the context of *prg-1* sterility when 22G RNAs are restored. I did not notice this reference or a comparison of the effects observed by Ketting and colleagues with the authors' observations.

We agree that there are some similarities between our model and the model proposed by the Ketting lab. However, the observations from the Ketting lab involved PRG-1's role in properly balancing HRDE-1 and CSR-1 22G accumulation in an environment where RdRP capacity is limited. The key observation was that without PRG-1 to properly set a proper boundary between targets which should be HRDE-1 dominant versus those that should be CSR-1 dominant, RdRP production between these competing factors become more evenly distributed, blurring the distinction between these competing Argonautes, leading to mis-regulation. Our observations suggest that even in the presence of PRG-1, the distinction between HRDE-1 and CSR-1 targets can breakdown. In fact, even though PRG-1 is dispersed into the cytoplasm in *glh-1/4* mutants, we saw that loss of *prg-1* in *glh-1/4* mutants suppressed mis-silencing, suggesting PRG-1 is responsible for triggering aberrant silencing in the absence of P granules. Our model therefore suggests the P granule environment allows PRG-1 and other Argonautes to work collectively to properly determine the expression or silencing of germline transcripts. We have now compared the mis-regulation found in *prg-1* to that found in VASA mutants.

5. What fraction of genes whose CSR-1-associated 22G RNAs are spread out along gene bodies remain expressed? If this is small, then this may be the sole decisive characteristic of CSR-1 targets that become silenced when GLH is mutant.

We thank the reviewer for this perspective, it caused us to look at the phenomenon from a larger perspective. We have identified two features of aberrantly silenced targets: these genes are more targeted by HRDE-1 in wild type animals, and these genes do not show the striking 3' enrichment of CSR-1 IP 22G-RNAs present for most CSR-1 targeted genes. We now characterized the extent to which these two features can predict the fate of mRNAs in *glh* mutants. However, neither of the features can confidently predict the fate of mRNAs in *glh* mutants. We have now add a few sentences and a figure (new Figure S6) to describe our findings.

First, we defined a set of CSR-1 targets that fail to show 3' end enrichment by selecting those with less than 15% of CSR-1 IP 22G-RNAs which map to the target mapping to the last 15% of the target transcript's length. Using this criterion, we obtained a list of 1905 CSR-1 targeted genes (representing 38.6% of CSR-1 transcripts), which distribute in the following pattern as a group:

We have also defined 572 CSR-1 genes (representing 11.6% of CSR-1 transcripts) that already show some favor to HRDE-1 targeting in the wild type background.

We wanted to know how many of the CSR-1 targets with enhanced HRDE-1 targeting in the *glh-1* mutant fall into either category. We saw that of the 320 CSR-1 targets with enhanced HRDE-1 targeting in the *glh-1* mutant, about two-thirds (215/320) have no 3' end enrichment in CSR-1 IP or show HRDE-1 favor in wild type animals (below). However, both features also identify many CSR-1 genes that did not meet the criteria of enhanced HRDE-1 targeted CSR-1 targets found in *glh-1* mutant.

We wondered how each of the two features, when considered on their own, could predict aberrant silencing in *glh* mutants. We saw that lack of 3' end enrichment and a HRDE-1 favored IP ratio both predict which transcripts will become aberrantly silenced (more HRDE-1 associated 22G-RNAs in *glh-1* mutants) better than the general classification of being a CSR-1 targeted gene, but neither category is distinct enough to fully predict which CSR-1 targets will become HRDE-1 targeted in P granule mutants:

We have added a discussion of the influence of these two features on aberrant silencing, and we have included the comparisons shown above to Figure S6.

6. For Piwi targets that become expressed, perhaps acknowledge that these may differ from those that remain silenced, and in the Discussion explicitly state that understanding this distinction remains an important question in the field. An alternative might be to offer a more wholistic model for how CSR and Piwi misregulation is coordinately achieved.

We thank the reviewer for pointing out this interesting perspective and we have now described our model in the discussion section now. It is reported that some piRNA targets, once silenced by piRNAs, can maintain silencing without its targeting piRNA for many generations. At the same time, other piRNA targets require PRG-1 and piRNAs to remain silenced by 22G-RNAs. Our analysis revealed that those genes which actively depend on PRG-1 for 22G accumulation (known as PRG-1-dependent piRNA targets) are the genes affected most by P granule loss (Figure S4F). Therefore, one model that could explain this correlation is that some PRG-1 targeted genes must be actively surveyed by PRG-1 each generation in P granules to be properly silenced. When P

granules are disrupted and PRG-1 is dispersed, it is these targets that are most affected. While this is one possibility, we agree that the true explanation remains unknown and is quite relevant for the field. (We have added the relevant discussion in the manuscript now.

7. The authors have made good progress by defining populations of RNA targets whose expression is modulated in response to P granule perturbation. If P granule disruption fails to affect expression of most Piwi and CSR-1 targets, perhaps there are small RNA amplification loops that occur in the nucleus or cytoplasm or in a residual P granule structure or in Z or Mutator granules. Even if one cannot detect the presence of a factor near the nucleus by microscopy, this factor could be there in small amounts. An open question may be precisely how the P granule disruption that the authors report affects the structure and function of P granules and associated bodies. Perhaps acknowledge this caveat in the Discussion.

We agree and have added this caveat to the discussion section.

8. The authors report that mutator foci are affected by GLH / Vasa but this is missing from the discussion. Mutator foci may be where 22G RNA biogenesis occurs, so perhaps it is alterations to these foci that are responsible for some alterations to 22G RNA populations. Do the authors feel that their RNA FISH clearly distinguishes localization to P granules rather than to Mutator foci? If so, why is the Discussion mostly focused on a role for P granules in their observations?

We thank the reviewer for pointing out the potential role of mutator foci.

One reason we think the defects stem from disruption of P granules is that the mutants we examined here, including GLHs, DEPS-1, PGL-1 or MIP-1/2 are all P granule factors known to be involved in P granule assembly. Importantly, while all these P granule assembly mutants all phenocopy *glh* mutants (exhibit mis-silencing and reduction or partial reduction of WAGO silencing), the mutator foci mutants (such as *mut-16*) or Z granule mutant (such as *ZNFX-1*) do not exhibit mis-silencing of CSR-1 targets but exhibit distinct small RNA defects, such as severe WAGO 22G synthesis defects and distribution changes of both WAGO and CSR-1 small RNAs. We suspect there may be a hierarchy in germ granule assembly and the defects of Z granule and mutator granule assembly that we report here stem from P granule assembly defects.

9. How does mRNA localization to P granules lead to 22G RNA biogenesis in mutator bodies that funnels 22G RNAs back to the P granule and to the correct Argonaute protein? This may not be well understood but is probably relevant to data in this paper. Do the authors imagine that CSR-1 associates with CSR-1 mRNA targets in P granules and that the 22G RNAs made from rare mRNAs in the mutator bodies then get funneled back into the P granule where the CSR-1-associated mRNAs soak up local concentrations of 22G RNAs into a CSR-1 sub-domain that might be critical for the 22G sorting process? If so, how does this tie into the altered 22G and mRNA expression

observed for a sub-set of Piwi an CSR targets? What is known about mRNA localization to Mutator bodies where RDRPs are concentrated? Some of these points might be offered in the Discussion to create a more coherent understanding of the framework of the problem being studied.

We thank the reviewer for raising these complex but interesting questions. We have now described a model in the discussion hopefully to create a more coherent understanding based on observations made in this manuscript and previous reports.

As *glh* mutants exhibit defects in not only P granule assembly, but also mutator and Z granule assembly, we do not believe our data can currently distinguish between these highly detailed and complex relationships within the germ granule environment, but we interpret our data to suggest that with this complex relationship disrupted, 22G-RNA accumulation does not cease but rather becomes discordant leading to propagated mis-regulation. We do favor the model that P granule assembly allows PIWI and CSR-1 to properly compete and identify their critical targets to achieve proper gene silencing and gene expression (as we discussed earlier that the defect of mis-regulation of both WAGO and CSR-1 targets can only be found in mutants defective in P granule assembly, but not Z granule or mutator granule).

Regarding CSR-1 22G-RNA synthesis, our model is that CSR-1 22G RNAs synthesis occurs in the cytoplasm and in the P granule, but not in the Mutator. The model is based on these previous studies: First, WAGO 22G-RNAs can be produced by RdRPs EGO-1 and RRF-1, while CSR-1 22G-RNAs are mainly made by EGO-1 (Gu et al., 2009). While RRF-1 co-localizes with MUT-16 in Mutator foci (Phillips et al 2012), EGO-1 seems to better co-localize with P granule factors (Claycomb et al 2009). Second, disruption of the Mutator by *mut-16* mutation grossly affects WAGO 22G-RNA but not CSR-1 production (Phillips et al 2012), (Gu et al 2009). Further, it has been recently argued by the Cecere lab that EGO-1 can function efficiently in the cytosol to produce CSR-1 22G-RNAs and this cytosolic pool may represent the main reservoir of CSR-1 22G-RNAs (Singh et al 2021).

10. page 9. These results indicate that localization of piRNA factors at perinuclear and cytoplasmic P granules can both contribute to their function in piRNA silencing. How do the authors data clearly show that cytoplasmic P granules promote silencing? Perhaps soften this conclusion?

By cytoplasmic P granules, we are referring to those cytoplasmic P granules observed in *C. elegans* embryos that have not yet become tethered to the nuclear periphery. It has been shown that *meg-3/4* mutants, which have defects in embryonic (cytoplasmic) P granule accumulation but not in adult (perinuclear) P granule accumulation, have defects in 22G-RNA accumulation which affect piRNA silencing and RNAi (Lev et al 2019). We found here that *meg-3/4* mutants also show piRNA silencing defects according to our piRNA reporter analysis. In addition, the *glh-1 FGGΔ* mutant which only shows adult perinuclear P granule dispersal also activated our piRNA reporter.

Therefore, we conclude that both cytoplasmic and perinuclear P granules contribute to piRNA silencing. We have now better clarified the nomenclature in the manuscript.

11. page 12. If *glh-1* single mutants are more compromised for disrupted 22G and mRNA levels, perhaps the authors should ask what characteristics are different or shared between *glh-1* single mutants and *glh-1 glh-4* mutants.

All our experiments indicate *glh-1 glh-4* mutants are more compromised in 22G-RNA production than *glh-1* single mutants. We only used *glh-1* single mutant to perform the HRDE-1 IP experiment purely out of technical limitations – we could not accumulate enough material for immunoprecipitation using the nearly sterile *glh-1 glh-4* double mutants. We have now added a few sentences in the results section to clarify this point.

We felt comfortable using the single mutant as a proxy because the phenotype of the single mutant is very similar to the double mutant in terms of granule dispersal and reporter activation, but just slightly less extreme. In fact, when we compare the small RNAs sequenced from single and double mutants, we see that while there is a very high degree of overlapping mis-regulation, the double mutants have more numerous WAGO targets with downregulated 22G-RNAs and more numerous CSR-1 targets with up-regulated 22G-RNAs (see below).

Minor:

1. 'In *C. elegans*, piRNAs and other small RNA pathways factors' – pathway factors
2. Page 6. Point out that the number of CSR-1 foci is similar but that the overall level of perinuclear CSR-1 is reduced.
3. line 317. In other mutants defecting in cytoplasmic and/or. defective
4. line 346. 'an aberrantly silenced CSR-1 mRNA, and noticed that *ceh-49* mRNAs were expressed'. an aberrantly silenced mRNA from a gene that is normally protected from silencing by CSR-1.

We thank the reviewer for the above comments – the text has been changed to reflect these corrections.

5. 'These results are consistent with the model that the P granule localization of some mRNA transcripts is critical for their protection from piRNA silencing.' If Piwi is also in P granules why are there more pro-silencing sRNAs made?

Our model to explain this phenomenon is that specific targets that are normally protected from pro-silencing sRNAs in the P granule lose that protection when granules are disrupted. We think this is likely due to the ability of PRG-1 to more easily find these mRNAs in the cytoplasm when CSR-1 protection is less localized. For a transcript like *ceh-49* which is protected from PRG-1 targeting in the wild type germline, CSR-1 is able to efficiently out-compete PRG-1 when PRG-1 is restricted to the P granule. However, when PRG-1 is dispersed into the cytoplasm, now CSR-1 cannot always reliably outcompete *ceh-49* for binding. Because PRG-1 can trigger pro-silencing sRNA production upon targeting an mRNA, transcripts like *ceh-49* which have this propensity to switch to becoming a silenced target are more likely to do so when PRG-1 is not confined to the P granule.

6. line 362. 'they are no longer distinct from WAGO genes'. that they no longer resemble WAGO targets?

We do mean that these enhanced HRDE-1 targeted CSR-1 genes are no longer distinguishable from WAGO genes. We have changed the language to more clearly reflect that meaning.

REVIEWERS' COMMENTS

Reviewer #1 (Remarks to the Author):

In this revised manuscript, the authors have adequately addressed all my comments and improved the manuscript's clarity.

Reviewer #2 (Remarks to the Author):

Authors successfully addressed all my previous concerns, thus, in my opinion the manuscript is ready for publication.

Reviewer #3 (Remarks to the Author):

Chen, Brown and Lee offer a revision of their manuscript on GLH function at P granules in promoting small RNA biogenesis. The authors have responded well to many comments raised by the reviewers. The results are explained much more clearly, and additional relevant material is now accurately integrated into the text. The authors nicely show that piRNA silencing defects can occur in many backgrounds where P granules are developmentally perturbed (either embryonic cytoplasmic P granules or perinuclear P granules of older embryos and larvae). Moreover, 22G silencing small RNAs corresponding to several hundred silenced genes decrease in *glh* mutants, and mRNA expression of these genes becomes activated. There is a corresponding >2-fold increase in 22G RNAs normally associated with CSR-1, which promotes gene expression, and these small RNAs are associated with the HRDE-1 silencing Argonaute and reduced mRNA expression.

The authors rigorously confirm reduced expression of these normally protected mRNAs in several genetic backgrounds with reduced P granule levels, including several genes required for fertility (notably possibly explaining why *glh-1 glh-4* mutants are barely fertile). Cytoplasmic P granules are less important for small RNA homeostasis, but these are only present during very early cell divisions of the embryo. These experiments define perinuclear P granules and related structures as important factors in promoting CSR-1 association with small RNAs that normally protect from silencing but can be misrouted to promote gene silencing. An excellent summary of how distinctions in self or non-self nucleic acids are made in P granules is provided, such that some PRG-1 silenced genes become expressed and some CSR-1 protected genes become silenced, in a manner that depends on mRNA localization to P granules. Somewhat puzzling is that piRNAs are required for the observed silencing, which does not suggest misrouting of effector 22G RNAs themselves. Instead, it is possible that altered creation of 22G RNAs by RDRPs in response to PRG-1 may be a central factor that is regulated by P granules, and that CSR-1 simply does a weaker job of stimulating 22G RNA production via EGO-1 for its mRNA targets when P granules are disrupted.

1. 'In addition, these abnormal, cytoplasmic aggregates are not properly sorted to the germ cell lineage, leading to the presence'

Alternatively, somatic cell degradation of P granules may not be functioning correctly (Seydoux paper)

2. The authors elegantly use genetics to demonstrate *prg-1* acts with piRNAs to promote aberrant silencing of mRNA transcripts that are normally protected by CSR-1 in the context of perinuclear P granules.

3. The authors now do a good job of acknowledging multiple interpretations of how changes in P granule size might lead to aberrantly high levels of siRNAs that associate with HRDE-1 silencing factor and promote inappropriately low mRNA levels.

4. RNA FISH probe sequences are not provided in the Methods.

REVIEWER COMMENTS

Reviewer #1 (Remarks to the Author):

The manuscript by Chen et al. describes the role of the two GLH/VASA helicases in germ granule formation and their impact on small RNA biosynthesis and function. They have evaluated several mutants of germ granule proteins and their effects on germ granule formations in adults and embryos. The removal of GLH-1/4 is sufficient to disrupt the organization of the three granule condensates (P and Z granules and mutator foci). Surprisingly, even though PRG-1 is destabilized by mutation of GLH-1/4, the piRNAs accumulation is only mildly perturbed. Instead, they observed more severe effects in the 22G-RNAs generated in mutator foci. They evaluated the loading of these 22G-RNAs in the nuclear Argonaute HRDE-1 upon mutation of GLH-1/4. They identified some reduced 22G-RNAs from mRNAs targeting piRNAs with a consequently mild increase in the corresponding mRNA accumulation. They have also identified the novel acquisition of 22G-RNAs from genes usually protected by the Argonaute CSR-1 (a small proportion though, 6%). They have observed a corresponding decrease in the accumulation of these mRNAs.

Interestingly, the CSR-1 protected genes targeted by HRDE-1 in GLH-1/4 already have some level of 22G-RNA loading in HRDE-1 compared to all the other CSR-1 targets. Therefore, the authors speculate that this might be the reason for the misrouting 22G-RNAs into HRDE-1 from CSR-1 22G-RNAs. In general, most of the experiments are well executed with proper controls, supporting the paper's conclusion. The findings presented are a bit descriptive, and no mechanistic insights are provided to elucidate how GLH-1/4 affects granule formations and small RNA biogenesis. Moreover, some of their findings corroborate previously published results, which should be acknowledged in the text.

We agree that it is interesting that GLH-1 is required for preventing mis-silencing of a subset of CSR-1 genes that seems to be silencing prone (targeted by both CSR-1 and surprisingly a bit by HRDE-1). We thank the reviewer for their insightful comments and their careful evaluation of our work. We also thank for the reviewer for pointing out previous work related to this study and we have now properly acknowledged their findings.

Main comments:

1. The authors should acknowledge the mass spec analysis of GLH-1 performed in Marnik et al. 2019. Here they have performed a more quantitative analysis (by LC-MS/MS) of the interactome of GLH-1 than the one presented by the authors in this manuscript and a previous manuscript (Chen et al. 2020). Most of the factors identified in this manuscript were also identified by Marnik et al. in a native condition of GLH-1 IPs. In addition, quantitative proteomic analysis of PRG-1 and CSR-1 complexes by Barucci et al. in a native condition also identified GLH-1 among the interacting proteins (especially in PRG-1 IPs where this interaction is quite strong and significant). Barucci

et al. also identified other granule components in their PRG-1 proteomic such as DEPS-1, WAGO-1, CSR-1, and Z granule factor WAGO-4. Therefore, the proteomic analysis and results presented in this manuscript are not novel and should be noted in the text.

We agree that the data presented in Marnik *et al* and in Barucci *et al* largely corroborate the evidence we present here from our own LC-MS/MS analysis, and we have now properly acknowledged their findings in our manuscript. While the experiment performed in Marnik *et al* using native GLH-1 IP is very similar to our LC-MS/MS experiment, here we additionally found evidence of interaction between GLH-1 and Z granule factor WAGO-4. The PRG-1 interactome reported in Barucci *et al* demonstrate an interaction between P, Mutator, and Z granule factors, and we have now acknowledged this important contribution in our manuscript.

2. On page 6, the authors wrote, “Surprisingly, the localization of MUT-16 was significantly disrupted in *glh-1 glh-4* double mutants (Figure 1D and S1C)”. I think it is worth mentioning that Singh et al. 2021 previously showed that RNAi of four P granule proteins (GLH-1, GLH-4, PGL-1, and PGL3) destabilize the formation of Z granule and the mutator foci and consequently the biogenesis of PRG-1-dependent 22G-RNAs. Similarly, on page 14, the authors wrote, “22G-RNAs antisense to CSR-1 target genes remain mostly unchanged in all mutants compared to wild type (Figure 5A, left and Figure S5A)”. Singh et al. have shown that CSR-1 22G-RNAs are produced in the cytosol, and removing the four P-granule proteins by RNAi does not change the abundance of CSR-1 22G-RNAs. The authors should mention these previous findings in the text, corroborating their results.

We thank the reviewer for pointing out this consistency with the previous report. We have now reported this connection and acknowledged the previous work in the text.

3. On page 7, the authors wrote, “In the *glh-1* DQAD mutant, large PRG-1 and WAGO-4 aggregates are found in the cytoplasm with a significant reduction in perinuclear PRG-1 and WAGO-4 foci (Figure 2B). In addition, these abnormal, cytoplasmic aggregates are not properly sorted to the germ cell lineage, leading to the presence of these foci in somatic lineages (Figure 2B)”. The characterization of the DQAD mutant and its effect on cytoplasmic granule aggregate in the germline and embryos has been previously shown by Marnik et al. 2019. They have also analyzed the impact of DQAD mutation on PGL-1 and PRG-1 localization. Therefore, the novel result here is the addition of WAGO-4 localization, which is not surprising given that proteomics data from the DQAD mutant protein in Marnik et al. also showed increased interaction with Argonaute proteins, such as CSR-1, PRG-1, WAGO-1, and C04F12.1. Moreover, previous results from the Lee lab also indicate the presence of large cytoplasmic aggregate in *glh-1* DQAD mutant adults and embryos, including GLH-1, PRG-1, CSR-1, and PGL-1 (Chen et al. 2020). Therefore, the results presented in this current manuscript should be considered in light of these previous publications.

We have now made mention of this previously published data as we agree our findings here are largely not novel. We agree that the WAGO-4 localization is not altogether

surprising given our own LC-MS/MS data showing WAGO-4 interaction with GLH-1.

4. In figure 4C, the authors should explain why in *meg-3/4* mutant, the mRNA of “reduced HRDE-1 targeting” and “other WAGO genes” are globally upregulated even though they have a mild reduction in corresponding 22G-RNAs. Also, maybe it would be better to remove data on DEPS-1 and PGL-1 since the authors do not show a complete dataset for these mutants (mRNAs and 22G-RNAs). The same comment is in Figure 5C.

We agree that the lack of a complete dataset for *deps-1* and *pgl-1* mutants makes these figures more difficult to interpret. We have removed the mRNA boxplots from these panels and moved mRNA expression data to the supplement. Additionally, we have expanded our analysis to include small RNA expression data from *mip-1/2* mutants, which have been shown to severely disrupt P granule integrity. We found that these mutants show very similar 22G-RNA defects, consistent with our model and granule disruption leads to subtle but specific mis-regulation.

The reviewer’s conclusion that all WAGO targeted mRNAs are globally upregulated despite a “mild” reduction in 22G-RNAs may have been due to our perhaps misleading representation of the data in Figure 4C. For the “Other WAGO genes” category, the median value for the log₂ 22G-RNA change in *meg-3/4* mutants was -1.09 compared to the log₂ mRNA change in *meg-3/4* mutants of 0.48. Therefore, the 22G-RNA changes were actually generally more extreme than the mRNA changes. This finding of profound 22G-RNA production defects in the *meg-3/4* mutant is consistent with previously published work from the Seydoux and Rechavi labs (Ouyang *et al* 2019 and Lev *et al* 2019, respectively). Because we needed to change the y-axis values in the 22G-RNA plots to accommodate the more extreme *glh-1* mutant datasets, a direct comparison between the mRNA and 22G-RNA boxplot panels was not possible. Due to our having now moved the mRNA expression panels to the supplement, we believe this confusion can now be avoided. Finally, the WAGO 22G-RNA changes in *meg-3/4* mutants is quite predictive of which mRNAs will show the most upregulation relative to wild type animals. 66% of WAGO targeted genes with reduced 22G-RNA expression show elevated mRNA expression:

5. In figure 2E, the authors show examples of smFISH aiming to demonstrate that the absence of GLH-1/4 affects the retention of the perinuclear smFISH signal compared to cytoplasmic smFISH detection. The authors should provide a quantification of these effects. Also, they might need to overlap the signal of smFISH with P-granule localization to make sure those perinuclear signals are mRNA molecules retained in germ granules.

We thank the reviewer for this suggestion. We were able to perform additional experiments which clarified our observation substantially. We tried two methods to quantify perinuclear localization, one based on the proximity of mRNA signal to DAPI (chromatin) signal and the other based on colocalization with P granule factor PRG-1. We found this second method to be more reliable.

In the first method, inferred from the DAPI staining where nuclei were positioned in 3D space, then measured the distance between mRNA foci and the inferred nuclear envelope. We found that this quantitation method was highly error-prone as we could not uniformly assign the perinuclear character of PRG-1, which is known to be highly enriched at the perinucleus. We think this difficulty comes from the uneven distribution of chromatin in germline nuclei, and the syncytial nature of the adult gonad which sometimes results in very crowded nuclei upon dissection and fixation. To accurately determine the true perinucleus, some nuclear membrane marker would be better suited and could be very useful in the future.

Simultaneously, we developed our smFISH protocol to be compatible with PRG-1 staining so that we could monitor P granule localization directly. We found that using our immunofluorescence-compatible smFISH protocol allowed us to measure the P granule

distribution of mRNA signal directly using colocalization with PRG-1. To perform this analysis properly, we used a version of the piRNA reporter strain that does not contain the silencing piRNA. We needed this additional strain to measure the effect of silencing on P granule colocalization because PRG-1 is dispersed in the *glh-1 glh-4* mutant, so it was not suitable to answer this question on its own. We have added this extensive analysis as Figure S2C. We found that when *gfp* mRNA is silenced, there is a significant increase in colocalization with PRG-1 protein. This is not the case for the germline expressed gene *nos-3*. Although the increase in colocalization was significant, the majority of the *gfp* mRNA signal in the silenced reporter strain does not colocalize with PRG-1. This could be due to partial colocalization with P granule adjacent bodies like the Mutator or Z granule. Because we only observed this modest effect, we discussed this caveat and softened our assertion that silenced *gfp* localizes to the perinucleus. We used the same protocol to directly test our assertion for the CSR-1 target that becomes silenced in P granule mutants – *ceh-49*. Similarly, we found that *ceh-49* mRNA does indeed show significantly more P granule accumulation than the germline expressed mRNA *nos-3*, but again the effect was mild. We have added this analysis as Figure S5C.

Reviewer #2 (Remarks to the Author):

Chen et al. in the study entitled ‘GLH/VASA helicases promote germ granule formation to ensure the fidelity of piRNA-mediated transcriptome surveillance’ performed an study to investigate if GLH/VASA helicase mutants have defects in forming perinuclear condensates containing PIWI and other small RNA cofactors. This investigation is performed in *Caenorhabditis elegans* worm, and expression of RNA was investigated. In parallel, a wide microscopy study was performed and mass spectrometry was used to investigate co-precipitated proteins. Regarding this latest technology, I’ve been contacted to review the proteomics part of this manuscript according to my expertise. The technology used and the mass spectrometry section of materials and methods is correct. LC-MSMS part is very detailed. But in my opinion, there are aspects that can be further improved, as for example the details about the in-gel digestion protocol.

We thank the reviewer for their diligence in reviewing our mass spectrometry data. We have expanded our description of the mass spectrometry methods, and we have addressed the comments point by point below. We believe that this important data in our paper is now much more understandable and will be easier for others to reproduce or reference as a result.

Additionally, a data analysis results section regarding how authors processed the proteomics data is totally missing. The number of replicates in the text is not stated. I realized that in the table contained in Figure 1 authors shows number of spectra detected in two replicates. This is also an important concern, regarding the validity of this data.

We have added a greatly expanded section in the methods detailing exactly how the proteomics data was processed. In this manuscript, we have done 2 biological replicates in our Mass spectrometry experiments, and the detailed protein identification results are provided in the supplementary table. The Mascot scores are provided in the supplemental table to reflect the confidence in protein identification.

Here I include a point-by-point revision of the proteomics-related aspects: Lines 309-311. When authors mention the immunoprecipitation study it is better to talk about proteins than genes.

Here, we are referring to the small RNAs sequencing data from immunoprecipitated protein complexes. Because these Argonaute protein complexes use these bound small RNAs to target mRNAs, we used this terminology to refer to the small RNA-targeted genes which share sequence complementarity with the sequenced protein bound small RNAs.

Section Mass spectrometry analysis. More details about in-gel digestion are needed. Otherwise, cite the reference used to follow the protocol of in-gel digestion if this protocol is already published in another article. Information regarding amount of trypsin used, if all the band was collected as a single sample for subsequent desalting, or gel was cut in "fractions" first, etc.

We have expanded our methods section to more explicitly detail the exact methodology used.

A section about data analysis for mass spectrometry is totally missing. It is not clear to me how authors can choose this such a small number of proteins of the immunoprecipitation study. Did authors obtained only 6 co-precipitated proteins together with glh-1? If more proteins were identified please specify how the selected proteins were chosen.

We obtained 231 and 242 co-precipitated proteins in GLH-1 IP replicates that passed the identification threshold of $FDR < 0.01$. This is compared to 30 and 11 co-precipitated proteins in the untagged control IP replicates. We selected the 6 proteins emphasized in Figure 1 because they are known factors of small RNA pathways. The full mass spectrometry dataset can be found in the supplemental table, which also contains Mascot scores to indicate statistical confidence in the correct protein identification.

Datasets with proteomics identifications are missing in supplementary.

We have supplied the full list of enriched protein identifications in the supplemental table. We hope they will be useful to other labs interested in GLH-1 and P granule complexes.

How many biological and technical replicates were performed in this proteomics study?
What kind of trypsin did authors use? Sequencing-grade?

We thank the reviewer for their attention to these essential details. We have added these to the methods. Briefly, two biological replicates were performed, and sequencing grade trypsin was used (Promega V5113).

Line 549. Remove “at”

Line 551. Rewrite the sentence in past tense.

Figure 1A. Did authors consider that two replicates are enough to obtain statistical significance in the proteomics study?

While more replicates will further increase our confidence, we found both replicates showed a highly similar set of identified proteins, and the several proteins emphasized in our study were never detected in untagged control IP experiments. As a consequence, we can be confident that these proteins are present in the GLH-1 immunoprecipitated complex. In addition, the identification of these factors passed the stringent statistical requirement imposed using MaxQuant software (PSM and protein FDR set at 0.01)

Reviewer #3 (Remarks to the Author):

22G RNAs are effector sRNAs whose structural properties appear identical but can promote very different outcomes: gene silencing or gene expression. A central puzzle in this regard concerns how sRNAs choose to or are chosen to associate with the CSR-1 anti-silencing Argonaute protein or the HRDE-1 pro-silencing Argonaute protein. Overall, this manuscript addresses this problem by showing that P granule structure may promote proper sorting of 22G small RNAs to CSR-1 via a population of target mRNAs that reside in P granules. Release of CSR-1-targeted mRNAs from P granules allows Piwi-associated piRNAs to create pro-silencing 22G RNAs that then associate with HRDE-1. One irony of these findings is that Piwi itself is concentrated in P granules and this association is profoundly disrupted by GLH Vasa dysfunction. Piwi may therefore be capable of promoting 22G RNA biogenesis at a location that is distinct from P granules, which is an unexpected discovery.

Chen and colleagues address a role for the germ granule protein Vasa in creating a perinuclear environment whose architecture is important for the association of small RNAs with the anti-silencing Argonaute CSR-1. In the absence of Vasa orthologues GLH-1 and GLH-4, a subset of genes that are protected from silencing by CSR-1-associated 22G RNAs become silenced. The mRNAs for some of these genes normally localize to P granules where they presumably associate with CSR-1 and may guide these mRNAs to mutator bodies where 22G RNAs are made that are transported back into P granules to associate with CSR-1 and then migrate in some fashion back into the nucleus to promote gene expression. In the absence of GLH/Vasa, a subset of CSR-1

target genes that normally display CSR-1-associated 22G RNAs across their gene bodies display reduced levels of transcription, and are targeted by HRDE-1-associated 22G RNAs that promote nuclear silencing. The CSR-1 targets that become silenced in the absence

of P granule proteins are in a 'mixed' category of genes that have 22G RNAs that associate with both HRDE-1 and CSR-1 in wildtype animals. Therefore, the category of CSR-1 targets that becomes silenced upon P granule dysfunction may be 'partially silenced' yet at the same time remains protected by CSR from silencing. A distinct category of CSR-1 targets remains protected from silencing by HRDE-1 and is enriched for 22G RNAs that map to the 3' UTRs of these genes. CSR-1 targets that remain protected by CSR-1 in the absence of GLH/Vasa might be consistent with weak but significant perinuclear localization of CSR-1.

Overall, this interesting manuscript reveals insight into how pro- and anti-silencing sRNAs targeting is shaped. Because the anti-silencing function of sRNAs remains a mysterious problem that is best understood in *C. elegans*, this manuscript offers insight that would be difficult or impossible to learn in another experimental system. The authors provide insight into how pro- and anti-silencing pathways are wired, and their unexpected discoveries represent a lot of work. A discussion of distinctions between genes whose expression is and is not affected by P granule function could more clearly convey what the authors have learned and what remains to be understood regarding how 22G RNAs that are structurally indistinguishable impart opposing effects at target loci within nuclei.

We thank the reviewer for their assessment of our manuscript and their constructive critiques that have undoubtedly improved our work. We have addressed individual comments below.

Comments:

1. Please move the model to a main figure, as this is essential for understanding the author's conclusions.

We have moved the model from the supplement to Figure 6.

2. A central conclusion is that CSR-1 function is more strongly tied to P granule structure for a subset of CSR-1 targets that is misregulated and normally has some HRDE-1-associated 22G RNAs. If 717 CSR-1 targets exhibited increased 22G RNA levels and 320 display HRDE-1-associated 22G RNA levels, then what about the other 400 CSR-1 targets? Do these genes display increased mRNA levels in *glh* mutants? Are these 22G RNAs associated with CSR-1 such that their numbers are increased in CSR-1 IP's? One reason that this might be important is that a steady-state level of 22G RNAs may be funneled to CSR-1, so if 320 genes have their 22G RNAs shifted to HRDE-1, then perhaps there is a corresponding increase in CSR-1-associated 22G RNAs for other target genes. These could be from natural CSR-1 targets or from Piwi targets that are no longer silenced.

One major reason for the drop from 717 aberrantly silenced CSR-1 targets in the total small RNA sequencing experiment to 320 targets showing increased HRDE-1 associated 22G-RNAs is that different requirements were set up for obtaining these genes; the 717 genes came from a calculation that only required a two-fold difference of 22G abundance between the mutant and wild type. For HRDE-1 IP-associated 22G-RNAs, we not only applied the two-fold cutoff, but also added a statistical cutoff of $p < 0.05$. We applied an additional cutoff for HRDE-1 IP since we sought to use this stricter gene list to better clarify precisely which genes were most affected by *glh* loss.

To examine whether those targets which have elevated 22G accumulation in *glh-1/4* mutants but did not meet the criteria of elevated HRDE-1 IP 22G-RNAs in *glh-1* mutants may have a distinct balance between HRDE-1 and CSR-1 IP, we remade Figure 5G using this group of genes (see below). However, we see a similar trend in the IP ratio, where this group has higher levels of HRDE-1/CSR-1 than other CSR-1 genes in the wild type background and this ratio further increases for the more stringent list, suggesting that these 600 CSR-1 targets that do not meet the statistical cutoff, they can still be considered part of this group of distinct aberrantly silenced targets that rely on germ granules for protection from silencing:

3. Discussion: piRNA silencing of non-self is reduced. Simultaneously, hundreds of self

RNAs are silenced'. It would be helpful if the authors could summarize in the Discussion how many CSR-1 targets do not change their expression when GLH is mutant relative to the fraction that is affected. This might convey a larger picture understanding how important Vasa or P granules are for CSR-1 function. If the majority of piRNA and CSR-1 targets remain unaltered, what does this mean about how P granules shape gene expression? What can be said about the rest of the targets? This discussion might allow the magnitude of the pro- and anti-silencing defects to be understood.

We thank the reviewer for raising these interesting points that help us better present our findings about the role of VASA and P granules in gene regulation in a larger picture. A relevant discussion has been now added in the discussion section.

There are ~ 6% (or 15% with less conservative criteria) of CSR-1 targets exhibiting increased HRDE-1 22G RNAs in VASA mutants, and 16% (or 41% with less conservative criteria) of WAGO targets exhibiting reduced HRDE-1 22G RNAs in VASA mutants. These results demonstrate that a significant portion of germline transcripts are mis-regulated by HRDE-1 22G-RNAs in VASA mutants. However, as both WAGO and CSR-1 22G-RNAs can establish epigenetic memories (Shirayama et al, 2012 and Conine et al., respectively), the silenced or expressed state of many germline transcripts may be preserved in the absence of P granules. In this model, those mRNAs which did not establish robust epigenetic memories would be those that exhibit more mis-regulation in P granule mutants. Indeed, we observed that PRG-1 dependent 22G-RNA targets, which depend on PRG-1/piRNAs at each generation to trigger gene silencing, are those which exhibit a greater reduction of 22G-RNAs on WAGO targets in *glh-1/4* mutants (Figure S4F).

In addition, a previous study from the Ketting lab has demonstrated that re-establishment of the 22G-RNA system in *prg-1* mutants leads to gene-mis-regulation in a stochastic manner that varies between worms⁴⁵. As our measurements of 22G-RNAs or mRNAs are from hundreds of thousands of worms, stochastic activation or silencing that may exist in individual worms may not be detected. Examining whether P granule mutant worms exhibit aberrant activation or silencing of germline transcripts stochastically will be interesting in future studies. Taken together, our current model is that P granules provide a critical environment that allows distinct Argonautes (including silencing PRG-1 and anti-silencing CSR-1 Argonautes) to survey their targets. Loss of P granules thus leads to a failure of Argonautes to properly identify their targets and mis-regulation of hundreds of germline targets results. Some other evidence that support this model are described in the responses below.

4. There is a model from the Ketting group that pro- and anti-silencing pathways become misregulated in the context of *prg-1* sterility when 22G RNAs are restored. I did not notice this reference or a comparison of the effects observed by Ketting and colleagues with the authors' observations.

We agree that there are some similarities between our model and the model proposed by the Ketting lab. However, the observations from the Ketting lab involved PRG-1's

role in properly balancing HRDE-1 and CSR-1 22G accumulation in an environment where RdRP capacity is limited. The key observation was that without PRG-1 to properly set a proper boundary between targets which should be HRDE-1 dominant versus those that should be CSR-1 dominant, RdRP production between these competing factors become more evenly distributed, blurring the distinction between these competing Argonautes, leading to mis-regulation. Our observations suggest that even in the presence of PRG-1, the distinction between HRDE-1 and CSR-1 targets can breakdown. In fact, even though PRG-1 is dispersed into the cytoplasm in *glh-1/4* mutants, we saw that loss of *prg-1* in *glh-1/4* mutants suppressed mis-silencing, suggesting PRG-1 is responsible for triggering aberrant silencing in the absence of P granules. Our model therefore suggests the P granule environment allows PRG-1 and other Argonautes to work collectively to properly determine the expression or silencing of germline transcripts. We have now compared the mis-regulation found in *prg-1* to that found in VASA mutants.

5. What fraction of genes whose CSR-1-associated 22G RNAs are spread out along gene bodies remain expressed? If this is small, then this may be the sole decisive characteristic of CSR-1 targets that become silenced when GLH is mutant.

We thank the reviewer for this perspective, it caused us to look at the phenomenon from a larger perspective. We have identified two features of aberrantly silenced targets: these genes are more targeted by HRDE-1 in wild type animals, and these genes do not show the striking 3' enrichment of CSR-1 IP 22G-RNAs present for most CSR-1 targeted genes. We now characterized the extent to which these two features can predict the fate of mRNAs in *glh* mutants. However, neither of the features can confidently predict the fate of mRNAs in *glh* mutants. We have now add a few sentences and a figure (new Figure S6) to describe our findings.

First, we defined a set of CSR-1 targets that fail to show 3' end enrichment by selecting those with less than 15% of CSR-1 IP 22G-RNAs which map to the target mapping to the last 15% of the target transcript's length. Using this criterion, we obtained a list of 1905 CSR-1 targeted genes (representing 38.6% of CSR-1 transcripts), which distribute in the following pattern as a group:

We have also defined 572 CSR-1 genes (representing 11.6% of CSR-1 transcripts) that already show some favor to HRDE-1 targeting in the wild type background.

We wanted to know how many of the CSR-1 targets with enhanced HRDE-1 targeting in the *glh-1* mutant fall into either category. We saw that of the 320 CSR-1 targets with enhanced HRDE-1 targeting in the *glh-1* mutant, about two-thirds (215/320) have no 3' end enrichment in CSR-1 IP or show HRDE-1 favor in wild type animals (below). However, both features also identify many CSR-1 genes that did not meet the criteria of enhanced HRDE-1 targeted CSR-1 targets found in *glh-1* mutant.

We wondered how each of the two features, when considered on their own, could predict aberrant silencing in *glh* mutants. We saw that lack of 3' end enrichment and a HRDE-1 favored IP ratio both predict which transcripts will become aberrantly silenced (more HRDE-1 associated 22G-RNAs in *glh-1* mutants) better than the general classification of being a CSR-1 targeted gene, but neither category is distinct enough to fully predict which CSR-1 targets will become HRDE-1 targeted in P granule mutants:

We have added a discussion of the influence of these two features on aberrant silencing, and we have included the comparisons shown above to Figure S6.

6. For Piwi targets that become expressed, perhaps acknowledge that these may differ from those that remain silenced, and in the Discussion explicitly state that understanding this distinction remains an important question in the field. An alternative might be to offer a more wholistic model for how CSR and Piwi misregulation is coordinately achieved.

We thank the reviewer for pointing out this interesting perspective and we have now described our model in the discussion section now. It is reported that some piRNA targets, once silenced by piRNAs, can maintain silencing without its targeting piRNA for many generations. At the same time, other piRNA targets require PRG-1 and piRNAs to remain silenced by 22G-RNAs. Our analysis revealed that those genes which actively depend on PRG-1 for 22G accumulation (known as PRG-1-dependent piRNA targets) are the genes affected most by P granule loss (Figure S4F). Therefore, one model that could explain this correlation is that some PRG-1 targeted genes must be actively surveyed by PRG-1 each generation in P granules to be properly silenced. When P

granules are disrupted and PRG-1 is dispersed, it is these targets that are most affected. While this is one possibility, we agree that the true explanation remains unknown and is quite relevant for the field. (We have added the relevant discussion in the manuscript now.

7. The authors have made good progress by defining populations of RNA targets whose expression is modulated in response to P granule perturbation. If P granule disruption fails to affect expression of most Piwi and CSR-1 targets, perhaps there are small RNA amplification loops that occur in the nucleus or cytoplasm or in a residual P granule structure or in Z or Mutator granules. Even if one cannot detect the presence of a factor near the nucleus by microscopy, this factor could be there in small amounts. An open question may be precisely how the P granule disruption that the authors report affects the structure and function of P granules and associated bodies. Perhaps acknowledge this caveat in the Discussion.

We agree and have added this caveat to the discussion section.

8. The authors report that mutator foci are affected by GLH / Vasa but this is missing from the discussion. Mutator foci may be where 22G RNA biogenesis occurs, so perhaps it is alterations to these foci that are responsible for some alterations to 22G RNA populations. Do the authors feel that their RNA FISH clearly distinguishes localization to P granules rather than to Mutator foci? If so, why is the Discussion mostly focused on a role for P granules in their observations?

We thank the reviewer for pointing out the potential role of mutator foci.

One reason we think the defects stem from disruption of P granules is that the mutants we examined here, including GLHs, DEPS-1, PGL-1 or MIP-1/2 are all P granule factors known to be involved in P granule assembly. Importantly, while all these P granule assembly mutants all phenocopy *glh* mutants (exhibit mis-silencing and reduction or partial reduction of WAGO silencing), the mutator foci mutants (such as *mut-16*) or Z granule mutant (such as *ZNFX-1*) do not exhibit mis-silencing of CSR-1 targets but exhibit distinct small RNA defects, such as severe WAGO 22G synthesis defects and distribution changes of both WAGO and CSR-1 small RNAs. We suspect there may be a hierarchy in germ granule assembly and the defects of Z granule and mutator granule assembly that we report here stem from P granule assembly defects.

9. How does mRNA localization to P granules lead to 22G RNA biogenesis in mutator bodies that funnels 22G RNAs back to the P granule and to the correct Argonaute protein? This may not be well understood but is probably relevant to data in this paper. Do the authors imagine that CSR-1 associates with CSR-1 mRNA targets in P granules and that the 22G RNAs made from rare mRNAs in the mutator bodies then get funneled back into the P granule where the CSR-1-associated mRNAs soak up local concentrations of 22G RNAs into a CSR-1 sub-domain that might be critical for the 22G sorting process? If so, how does this tie into the altered 22G and mRNA expression

observed for a sub-set of Piwi an CSR targets? What is known about mRNA localization to Mutator bodies where RDRPs are concentrated? Some of these points might be offered in the Discussion to create a more coherent understanding of the framework of the problem being studied.

We thank the reviewer for raising these complex but interesting questions. We have now described a model in the discussion hopefully to create a more coherent understanding based on observations made in this manuscript and previous reports.

As *glh* mutants exhibit defects in not only P granule assembly, but also mutator and Z granule assembly, we do not believe our data can currently distinguish between these highly detailed and complex relationships within the germ granule environment, but we interpret our data to suggest that with this complex relationship disrupted, 22G-RNA accumulation does not cease but rather becomes discordant leading to propagated mis-regulation. We do favor the model that P granule assembly allows PIWI and CSR-1 to properly compete and identify their critical targets to achieve proper gene silencing and gene expression (as we discussed earlier that the defect of mis-regulation of both WAGO and CSR-1 targets can only be found in mutants defective in P granule assembly, but not Z granule or mutator granule).

Regarding CSR-1 22G-RNA synthesis, our model is that CSR-1 22G RNAs synthesis occurs in the cytoplasm and in the P granule, but not in the Mutator. The model is based on these previous studies: First, WAGO 22G-RNAs can be produced by RdRPs EGO-1 and RRF-1, while CSR-1 22G-RNAs are mainly made by EGO-1 (Gu et al., 2009). While RRF-1 co-localizes with MUT-16 in Mutator foci (Phillips et al 2012), EGO-1 seems to better co-localize with P granule factors (Claycomb et al 2009). Second, disruption of the Mutator by *mut-16* mutation grossly affects WAGO 22G-RNA but not CSR-1 production (Phillips et al 2012), (Gu et al 2009). Further, it has been recently argued by the Cecere lab that EGO-1 can function efficiently in the cytosol to produce CSR-1 22G-RNAs and this cytosolic pool may represent the main reservoir of CSR-1 22G-RNAs (Singh et al 2021).

10. page 9. These results indicate that localization of piRNA factors at perinuclear and cytoplasmic P granules can both contribute to their function in piRNA silencing. How do the authors data clearly show that cytoplasmic P granules promote silencing? Perhaps soften this conclusion?

By cytoplasmic P granules, we are referring to those cytoplasmic P granules observed in *C. elegans* embryos that have not yet become tethered to the nuclear periphery. It has been shown that *meg-3/4* mutants, which have defects in embryonic (cytoplasmic) P granule accumulation but not in adult (perinuclear) P granule accumulation, have defects in 22G-RNA accumulation which affect piRNA silencing and RNAi (Lev et al 2019). We found here that *meg-3/4* mutants also show piRNA silencing defects according to our piRNA reporter analysis. In addition, the *glh-1 FGGΔ* mutant which only shows adult perinuclear P granule dispersal also activated our piRNA reporter.

Therefore, we conclude that both cytoplasmic and perinuclear P granules contribute to piRNA silencing. We have now better clarified the nomenclature in the manuscript.

11. page 12. If *glh-1* single mutants are more compromised for disrupted 22G and mRNA levels, perhaps the authors should ask what characteristics are different or shared between *glh-1* single mutants and *glh-1 glh-4* mutants.

All our experiments indicate *glh-1 glh-4* mutants are more compromised in 22G-RNA production than *glh-1* single mutants. We only used *glh-1* single mutant to perform the HRDE-1 IP experiment purely out of technical limitations – we could not accumulate enough material for immunoprecipitation using the nearly sterile *glh-1 glh-4* double mutants. We have now added a few sentences in the results section to clarify this point.

We felt comfortable using the single mutant as a proxy because the phenotype of the single mutant is very similar to the double mutant in terms of granule dispersal and reporter activation, but just slightly less extreme. In fact, when we compare the small RNAs sequenced from single and double mutants, we see that while there is a very high degree of overlapping mis-regulation, the double mutants have more numerous WAGO targets with downregulated 22G-RNAs and more numerous CSR-1 targets with up-regulated 22G-RNAs (see below).

Minor:

1. 'In C. elegans, piRNAs and other small RNA pathways factors' – pathway factors
2. Page 6. Point out that the number of CSR-1 foci is similar but that the overall level of perinuclear CSR-1 is reduced.
3. line 317. In other mutants defecting in cytoplasmic and/or. defective
4. line 346. 'an aberrantly silenced CSR-1 mRNA, and noticed that ceh-49 mRNAs were expressed'. an aberrantly silenced mRNA from a gene that is normally protected from silencing by CSR-1.

We thank the reviewer for the above comments – the text has been changed to reflect these corrections.

5. 'These results are consistent with the model that the P granule localization of some mRNA transcripts is critical for their protection from piRNA silencing.' If Piwi is also in P granules why are there more pro-silencing sRNAs made?

Our model to explain this phenomenon is that specific targets that are normally protected from pro-silencing sRNAs in the P granule lose that protection when granules are disrupted. We think this is likely due to the ability of PRG-1 to more easily find these mRNAs in the cytoplasm when CSR-1 protection is less localized. For a transcript like *ceh-49* which is protected from PRG-1 targeting in the wild type germline, CSR-1 is able to efficiently out-compete PRG-1 when PRG-1 is restricted to the P granule. However, when PRG-1 is dispersed into the cytoplasm, now CSR-1 cannot always reliably outcompete *ceh-49* for binding. Because PRG-1 can trigger pro-silencing sRNA production upon targeting an mRNA, transcripts like *ceh-49* which have this propensity to switch to becoming a silenced target are more likely to do so when PRG-1 is not confined to the P granule.

6. line 362. 'they are no longer distinct from WAGO genes'. that they no longer resemble WAGO targets?

We do mean that these enhanced HRDE-1 targeted CSR-1 genes are no longer distinguishable from WAGO genes. We have changed the language to more clearly reflect that meaning.

REVIEWERS' COMMENTS

Reviewer #1 (Remarks to the Author):

In this revised manuscript, the authors have adequately addressed all my comments and improved the manuscript's clarity.

We thank the reviewer for their helpful comments, they have undoubtedly improved our manuscript.

Reviewer #2 (Remarks to the Author):

Authors successfully addressed all my previous concerns, thus, in my opinion the manuscript is ready for publication.

We thank the reviewer for their diligence. Our discussion of the proteomics in our manuscript has improved significantly.

Reviewer #3 (Remarks to the Author):

Chen, Brown and Lee offer a revision of their manuscript on GLH function at P granules in promoting small RNA biogenesis. The authors have responded well to many comments raised by the reviewers. The results are explained much more clearly, and additional relevant material is now accurately integrated into the text. The authors nicely show that piRNA silencing defects can occur in many backgrounds where P granules are developmentally perturbed (either embryonic cytoplasmic P granules or perinuclear P granules of older embryos and larvae). Moreover, 22G silencing small RNAs corresponding to several hundred silenced genes decrease in *glh* mutants, and mRNA expression of these genes becomes activated. There is a corresponding >2-fold increase in 22G RNAs normally associated with CSR-1, which promotes gene expression, and these small RNAs are associated with the HRDE-1 silencing Argonaute and reduced mRNA expression.

The authors rigorously confirm reduced expression of these normally protected mRNAs in several genetic backgrounds with reduced P granule levels, including several genes required for fertility (notably possibly explaining why *glh-1 glh-4* mutants are barely fertile). Cytoplasmic P granules are less important for small RNA homeostasis, but these are only present during very early cell divisions of the embryo. These experiments define perinuclear P granules and related structures as important factors in promoting CSR-1 association with small RNAs that normally protect from silencing but can be misrouted to promote gene silencing. An excellent summary of how distinctions in self or non-self nucleic acids are made in P granules is provided, such that some PRG-1 silenced genes become expressed and some CSR-1 protected genes become silenced, in a manner that depends on mRNA localization to P granules. Somewhat puzzling is that piRNAs are required for the observed silencing, which does not suggest mis-routing of effector 22G RNAs themselves. Instead, it is possible that altered creation of 22G RNAs by RDRPs in response to PRG-1 may be a central factor

that is regulated by P granules, and that CSR-1 simply does a weaker job of stimulating 22G RNA production via EGO-1 for its mRNA targets when P granules are disrupted.

1. 'In addition, these abnormal, cytoplasmic aggregates are not properly sorted to the germ cell lineage, leading to the presence'

Alternatively, somatic cell degradation of P granules may not be functioning correctly (Seydoux paper)

We agree that this alternative explanation is also possible. We have added this alternative to our manuscript.

2. The authors elegantly use genetics to demonstrate prg-1 acts with piRNAs to promote aberrant silencing of mRNA transcripts that are normally protected by CSR-1 in the context of perinuclear P granules.

We thank the reviewer and agree.

3. The authors now do a good job of acknowledging multiple interpretations of how changes in P granule size might lead to aberrantly high levels of siRNAs that associate with HRDE-1 silencing factor and promote inappropriately low mRNA levels.

We thank the reviewer and agree.

4. RNA FISH probe sequences are not provided in the Methods.

We have now added all smFISH probe sequences used in this study in the supplementary material.